# Functional Complexity-adaptive Temporal Tensor Decomposition

**Panqi Chen**[1]    **Lei Cheng**[1, 2] *    **Jianlong Li**[1]    **Weichang Li**[1]
**Weiqing Liu**[3]    **Jiang Bian**[3]    **Shikai Fang**[1, 3]*
[1]College of Information Science and Electronic Engineering, Zhejiang University
[2]Zhejiang Provincial Key Laboratory of
Multi-Modal Communication Networks and Intelligent Information Processing
[3]Microsoft Research Asia

## Abstract

Tensor decomposition is a fundamental tool for analyzing multi-dimensional data by learning low-rank factors to represent high-order interactions. While recent works on temporal tensor decomposition have made significant progress by incorporating continuous timestamps in latent factors, they still struggle with general tensor data with continuous indexes not only in the temporal mode but also in other modes, such as spatial coordinates in climate data. Moreover, the challenge of self-adapting model complexity is largely unexplored in functional temporal tensor models, with existing methods being inapplicable in this setting. To address these limitations, we propose functional **C**omplexity-**A**daptive **T**emporal **T**ensor d**E**composition (CATTE). Our approach encodes continuous spatial indexes as learnable Fourier features and employs neural ODEs in latent space to learn the temporal trajectories of factors. To enable automatic adaptation of model complexity, we introduce a sparsity-inducing prior over the factor trajectories. We develop an efficient variational inference scheme with an analytical evidence lower bound, enabling sampling-free optimization. Through extensive experiments on both synthetic and real-world datasets, we demonstrate that CATTE not only reveals the underlying ranks of functional temporal tensors but also significantly outperforms existing methods in prediction performance and robustness against noise. The code is available at `https://github.com/OceanSTARLab/CATTE`.

## 1   Introduction

Tensor is a ubiquitous data structure for organizing multi-dimensional data. For example, a four-mode tensor *(longitude, latitude, depth, time)* can serve as a unified representation of spatiotemporal signals in the ocean, such as temperature or flow speed. Tensor decomposition is a prevailing framework for multiway data analysis that estimates latent factors to reconstruct the unobserved entries. Methods like CANDECOMP/PARAFAC (CP)[1] and Tucker decomposition[2] are widely applied across fields, including climate science, oceanography, and social science.

An emerging trend in tensor community is to leverage the continuous timestamp of observed entries and build temporal tensor models, as the real-world tensor data is often irregularly collected in time, e.g., physical signals, accompanied with rich and complex time-varying patterns. The temporal tensor methods expand the classical tensor framework by using polynomial splines [3], Gaussian processes [4, 5], ODE [6] and energy-based models [7] to estimate the continuous temporal dynamics in latent space, instead of discretizing the time mode and setting a fixed number of factors.

---

*Correspond to lei_cheng@zju.edu.cn, fsk@zju.edu.cn

Despite the successes of current temporal tensor methods, they inherit a fundamental limitation from traditional tensor models: they assume tensor data at each timestep must conform to a Cartesian grid structure with discrete indexes and finite-dimensional modes. This assumption poorly aligns with many real-world scenarios where modes are naturally continuous, such as spatial coordinates like *(longitude, latitude, depth)*. To fit current models, we still need to discretize continuous indexes, which inevitably leads to a loss of fine-grained information encoded in these indexes. From a high-level perspective, while current temporal tensor methods have taken a crucial step forward by modeling continuous characteristics in the temporal mode compared to classical approaches, they still fail to fully utilize the rich complex patterns inherent in other continuous-indexed modes.

Another crucial challenge lies in automatically adapting the complexity of functional temporal tensor decomposition model to the data, which is determined by the tensor rank. As a key hyperparameter in tensor modeling, the rank directly influences interpretability, sparsity, and model expressiveness. While classical tensor literature offers extensive theoretical analysis and learning-based solutions [8, 9, 10], this topic has been largely overlooked in emerging functional temporal tensor methods. The introduction of dynamical patterns significantly complicates the latent landscape, and the lack of investigation into rank selection makes temporal tensor models more susceptible to hyperparameter choice. Exisiting approaches [8, 9, 10] focus on modeling discrete factors, which can not be straightforwardly extended to the functional regimes.

To fill these gaps, we propose CATTE, a *complexity-adaptive* method for modeling temporal tensor data with *continuous indexes across all modes*. Our method models general temporal tensor data with continuous indexes not only in the time mode but also in other modes. Specifically, CATTE organizes the continuous indexes from non-temporal modes into a Fourier-feature format, encodes them as the initial state of latent dynamics, and utilizes the neural ODE [11] to model the factor trajectories. To enable self-adaptive complexity in functional temporal tensor decomposition, CATTE extends the classical Bayesian rank selection framework [8] by placing sparsity-inducing priors over *factor trajectories*. For efficient inference, we propose a novel variational inference algorithm with an analytical evidence lower bound, enabling *sampling-free inference* of the model parameters and latent dynamics. For evaluation, we conducted experiments on both simulated and real-world tasks, demonstrating that CATTE not only reveals the underlying ranks of functional temporal tensors but also significantly outperforms existing methods in prediction performance and robustness against noise.

## 2   Preliminary

### 2.1   Tensor Decomposition

Tensor decomposition represents multi-dimensional arrays by decomposing them into lower-dimensional components, thus revealing underlying patterns in high-dimensional data. We denote a $K$-mode tensor as $\mathcal{Y} \in \mathbb{R}^{I_1 \times \cdots \times I_k \times \cdots \times I_K}$, where the $k$-$th$ mode consists of $I_k$ dimensions. Each entry of $\mathcal{Y}$, termed $y_\mathbf{i}$, is indexed by a $K$-tuple $\mathbf{i} = (i_1, \cdots, i_k, \cdots, i_K)$, where $i_k$ denotes the index of the node along the mode $k$ ($1 \le k \le K$). For tensor decomposition, a set of factor matrices $\{\mathbf{U}^k\}_{k=1}^K$ are introduced to represent the nodes in each mode. Specifically, the $k$-$th$ factor matrix $\mathbf{U}^k$ is composed of $I_k$ latent factors, i.e., $\mathbf{U}^k = [\mathbf{u}_1^k, \cdots, \mathbf{u}_{i_k}^k, \cdots, \mathbf{u}_{I_k}^k]^\mathrm{T} \in \mathbb{R}^{I_k \times R_k}$ and $\mathbf{u}_{i_k}^k = [u_{i_k,1}^k, \cdots, u_{i_k,r_k}^k, \cdots, u_{i_k,R_k}^k]^\mathrm{T} \in \mathbb{R}^{R_k}$, where $R_k$ denotes the rank of mode-$k$. The classic CANDECOMP/PARAFAC (CP) decomposition [1] aims to decompose a tensor into a sum of rank-one tensors. It sets $R_1 = \cdots = R_k = \cdots = R_K = R$ and represents each entry using

$$y_\mathbf{i} \approx \mathbf{1}^\mathrm{T}[\underset{k}{\circledast}\mathbf{u}_{i_k}^k] = \sum_{r=1}^{R}\prod_{k=1}^{K} u_{i_k,r}^k, \tag{1}$$

where $\mathbf{1} \in \mathbb{R}^R$ is the all-one vector and $\underset{k}{\circledast}$ is Hadamard product of a set of vectors defined as $\underset{k}{\circledast}\mathbf{u}_{i_k}^k = (\mathbf{u}_{i_1}^1 \circledast \cdots \circledast \mathbf{u}_{i_k}^k \circledast \cdots \circledast \mathbf{u}_{i_K}^K)$. Here, $\circledast$ is Hadamard product.

### 2.2   Automatic Tensor Rank Determination

The tensor rank $R$ determines the complexity of the tensor model. An improper choice of the rank can lead to overfitting or underfitting to the signal sources, potentially compromising the model

interpretability. However, the optimal determination of the tensor rank is known to be NP-hard [9, 12, 13]. Bayesian methods have been introduced to facilitate Tucker/CP decomposition with automatic tensor rank learning [14, 8, 9, 10]. These methods impose sparsity-promoting priors (e.g., the Gaussian-Gamma prior and Laplacian prior) on the latent factors.

For example, Bayesian CP decomposition with Gaussian-Gamma priors models the mean and precision of all latent factors with zero elements and a set of latent variables $\boldsymbol{\lambda} = [\lambda_1, \cdots, \lambda_r, \cdots, \lambda_R]^{\mathrm{T}} \in \mathbb{R}^R$, respectively:

$$p(\mathbf{u}_{i_k}^k|\boldsymbol{\lambda}) = \mathcal{N}(\mathbf{u}_{i_k}^k|\mathbf{0}, \boldsymbol{\Lambda}^{-1}), \forall k, \quad p(\boldsymbol{\lambda}) = \prod_{r=1}^{R} \mathrm{Gamma}(\lambda_r|a_r^0, b_r^0), \tag{2}$$

where $\boldsymbol{\Lambda} = \mathrm{diag}(\boldsymbol{\lambda})$ is the inverse covariance matrix shared by all latent factors over $K$ modes. Note that $R$ components of $\mathbf{u}_{i_k}^k$ are assumed to be statistically independent and the distribution of the $r$-$th$ component is controlled by $\lambda_r$. For example, if $\lambda_r$ is large, then the density function peaks at mean zero, so the $r$-$th$ component is concentrated at zero. Otherwise, if $\lambda_r$ is small (which leads to heavy tails), it allows the component to spread out to wider range of values. The conjugated Gamma priors are assigned to $\boldsymbol{\lambda}$. Here $\mathrm{Gamma}(x|a, b) = \frac{b^a x^{a-1} e^{-bx}}{\Gamma(a)}$ for $x \geq 0$, which represents the Gamma distribution for $\boldsymbol{\lambda}$. In this context, $a$ and $b$ denote the shape and rate parameters respectively, and $\Gamma(\cdot)$ denotes the Gamma function. $\{a_r^0, b_r^0\}_{r=1}^{R}$ are pre-determined hyperparameters. The tensor rank will be automatically determined by the inferred posteriors of $\boldsymbol{\lambda}$.

## 2.3 Generalized Tensor with Continuous Modes

Real-world tensor data often contains continuous modes, prompting increased studies on generalized tensor data with continuous indexes. Existing approaches can be broadly classified into two categories:

*1.Temporal tensor model with continuous timestamps:* Recent studies encode the representations of continuous timestamps into the latent factor of CP model [5, 15] or the tensor core of Tucker model [4]. Taking CP as an example, the temporal tensor model with continuous timestamps can be written as:

$$y_{\mathbf{i}}(t) \approx \mathbf{1}^{\mathrm{T}}[\underset{k}{\circledast} \mathbf{u}_{i_k}^k(t)], \tag{3}$$

where $\mathbf{i}$ is the tensor index, $t$ is the continuous timestamp, $\mathbf{u}_{i_k}^k(t)$ is the factor trajectory of the $i_k$-th node in the $k$-th mode. Although this modeling approach effectively captures complex temporal dynamics, it is inadequate for generalizing to data with continuous indexes over the entire domain, such as spatiotemporal data which has continuous coordinates on both spatial and temporal modes [16].

*2.Functional tensor model:* Another popular model to handle continuous-indexed modes is the functional tensor [17, 18, 19], which assumes that the continuous-indexed tensor can be factorized as a set of mode-wise functions and the continuous timestamp is simply modeled as an extra mode. Still taking CP as an example, the functional tensor model can be written as:

$$y_{\mathbf{i}}(t) \approx \mathbf{1}^{\mathrm{T}}[\underset{k}{\circledast} \mathbf{u}^k(i_k) \circledast \mathbf{u}^{\mathrm{Temporal}}(t)], \tag{4}$$

where $\mathbf{u}^k(i_k) : \mathbb{R}_+ \to \mathbb{R}^R$ is the latent vector-valued function of the $k$-th mode, which takes the continuous index $i_k$ as input and outputs the latent factor. $\mathbf{u}^{\mathrm{Temporal}}(t) : \mathbb{R}_+ \to \mathbb{R}^R$ is the latent function of the temporal mode. The fully-factorized form of (4) models each mode equally and independently. *It often overlooks the complex dynamics of the temporal mode, which requires special treatment [16].*

## 3 Methodology

Despite recent advances in modeling temporal tensors, most of these methods are still unsuitable for generalized tensor data with continuous indexes across all domains. While functional tensor methods offer greater flexibility, they simply treat temporal dynamics as an independent mode, which tends to underfit the inherent complexity of the temporal dynamics. Furthermore, rank determination remains a less explored issue in temporal tensor models. To address these issues, we propose

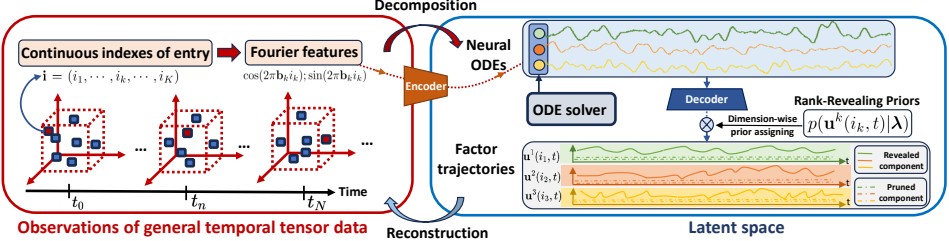

Figure 1: Graphical illustration of the proposed CATTE (the case of $K = 3$).

CATTE, a novel temporal tensor model that integrates the continuous-indexed features into a latent ODE model with rank-revealing prior.

Without loss of generality, we consider a $K$-mode generalized temporal tensor with continuous indexes over all domains, and it actually corresponds to a function $\mathbf{F}(i_1, \cdots, i_K, t) : \mathbb{R}_+^{K+1} \to \mathbb{R}^1$ to map the continuous indexes and timestamp to the tensor entry, denoted as $y_{\mathbf{i}}(t) = \mathbf{F}(i_1, \cdots, i_K, t)$. We assume the function $\mathbf{F}(i_1, \cdots, i_K, t)$ can be factorized into $K$ factor trajectories following the CP format with rank $R$, i.e.,

$$y_{\mathbf{i}}(t) = \mathbf{F}(i_1, \cdots, i_K, t) \approx \mathbf{1}^{\mathrm{T}}[\circledast_k \mathbf{u}^k(i_k, t)], \tag{5}$$

where $\mathbf{u}^k(i_k, t) : \mathbb{R}_+^2 \to \mathbb{R}^R$ is the trajectory of latent factor, which is a $R$-size vector-valued function mapping the continuous index $i_k$ of mode-$k$ and timestamp $t$ to a $R$-dimensional latent factor. We claim that the proposed model (5) is a generalization of existing temporal tensor methods (3) via modeling continuous-indexed patterns not only in the temporal mode but in all modes. If we restrict $i_k$ to finite and discrete, (5) will degrade to (3). Compared to the fully-factorized functional tensor (4), the proposed method (5) explicitly models the time-varying factor trajectories of all modes, known as dynamic factor learning [5, 15]. Given the fact that temporal mode always dominates and interacts with other modes, the proposed method is expected to improve the model's capability by learning time-varying representations in dynamical data.

### 3.1 Continuous-indexed Latent-ODE

To allow flexible modeling of the factor trajectory and continuous-indexed information, we propose a temporal function $\mathbf{g}^k(i_k, t) : \mathbb{R}_+^2 \to \mathbb{R}^R$ based on encoder-decoder structure and neural ODE [11] to approximate the factor trajectory $\mathbf{u}^k(i_k, t)$ of mode-$k$. Specifically, we have:

$$\mathbf{z}^k(i_k, 0) = \text{Encoder}\big([\cos(2\pi\mathbf{b}_k i_k); \sin(2\pi\mathbf{b}_k i_k)]\big), \tag{6}$$

$$\mathbf{z}^k(i_k, t) = \mathbf{z}^k(i_k, 0) + \int_0^t h_{\boldsymbol{\theta}_k}(\mathbf{z}^k(i_k, s), s)ds, \tag{7}$$

$$\mathbf{g}^k(i_k, t) = \text{Decoder}\big(\mathbf{z}^k(i_k, t)\big). \tag{8}$$

Eq. (6) shows that how we obtain $\mathbf{z}(i, 0) \in \mathbb{R}^J$, the initial state of the latent dynamics by encoding the continuous index $i_k$. In particular, the input coordinate $i_k$ is firstly expanded into a set of Fourier features $[\cos(2\pi\mathbf{b}_k i_k); \sin(2\pi\mathbf{b}_k i_k)] \in \mathbb{R}^{2M}$, where $\mathbf{b}_k \in \mathbb{R}^M$ is a learnable vector that scales $i_k$ by $M$ different frequencies. This effectively expands the input space with high-frequency components [20], helping to capture fine-grained index information. The Fourier features are then fed into an encoder $\text{Encoder}(\cdot) : \mathbb{R}^{2M} \to \mathbb{R}^J$ to get $\mathbf{z}^k(i_k, 0)$. Give the initial state, we then apply a neural network $h_{\boldsymbol{\theta}_k}(\mathbf{z}^k(i_k, s), s) : \mathbb{R}^J \to \mathbb{R}^J$ to model the state transition of the dynamics at each timestamp, which is parameterized by $\boldsymbol{\theta}_k$, and the state value can be calculated through integrations as shown in (7). Finally, we will pass the output of the latent dynamics through a decoder $\text{Decoder}(\cdot) : \mathbb{R}^J \to \mathbb{R}^R$ to obtain $\mathbf{g}^k(i_k, t)$ as the approximation of the factor trajectory, as described in (8). We simply use the multilayer perceptrons (MLPs) to parameterize the encoder and the decoder.

Note that (7) actually represents the neural ODE model [21, 11], and we handle the integration to obtain $\mathbf{z}^k(i_k, t)$ in (7) at arbitrary $t$ by using numerical ODE solvers:

$$\mathbf{z}^k(i_k, t) = \text{ODESolve}(\mathbf{z}^k(i_k, 0), h_{\boldsymbol{\theta}_k}). \tag{9}$$

**Algorithm 1:** Training process of CATTE

---

**Input:** Training data $\mathcal{D} = \{y_n, \mathbf{i}_n, t_n\}_{n=1}^N$
Collect all possible $I_k$ indexes for $K$ modes and $T$ possible timestamps. Initialize
$\quad \{\boldsymbol{\omega}_k\}_{k=1}^K, \{\alpha_r, \beta_r\}_{r=1}^R, \sigma, \rho, \iota.$
**while** *not convergence* **do**
$\quad$ Construct a set of initial ODE state tables $\mathcal{Z}_0$ using Fourier features and Encoder,
$\quad\quad$ encompassing all possible indexes.
$\quad$ **for** $i = 1, 2, \cdots, T$ **do**
$\quad\quad$ $\mathcal{Z}(t_i) = \text{ODESolve}(\mathcal{Z}(t_{i-1}), \{h_{\boldsymbol{\theta}_k}\}_{k=1}^K, (t_{i-1}, t_i))$
$\quad\quad$ Compute necessary $\mathbf{g}^k(i_k, t_i)$ from $\mathcal{Z}(t)$ using (8).
$\quad$ **end**
$\quad$ Take gradient step on (19).
**end**

---

For computing efficiency, we concatenate the initial states of multiple indexes together, and construct a matrix-valued trajectory, where each row corresponds to the initial state of an unique index. Then, we only need to call the ODE solver once to obtain the factor trajectory of observed indexes. For simplicity, we denote the all learnable parameters in (6)-(8) as $\boldsymbol{\omega}_k$ for mode-$k$, which includes the frequency-scale vectors $\mathbf{b}_k$ as well as the parameters of neural ODE $\boldsymbol{\theta}_k$ and the encoder-decoder.

### 3.2 Functional Rank-revealing Prior over Factor Trajectories

To automatically determine the underlying rank in the functional dynamical scenario, we apply the Bayesian sparsity-promoting priors. Different from the classical framework [8], stated in (2), we assign a dimension-wise Gaussian-Gamma prior to the factor trajectory,

$$p(\mathbf{u}^k(i_k, t)|\boldsymbol{\lambda}) = \mathcal{N}(\mathbf{u}^k(i_k, t)|\mathbf{0}, \boldsymbol{\Lambda}^{-1}), \forall k, \tag{10}$$

where $\boldsymbol{\Lambda} = \text{diag}(\boldsymbol{\lambda})$ and $\boldsymbol{\lambda} = [\lambda_1, \cdots, \lambda_r, \cdots, \lambda_R]^T \in \mathbb{R}^R$. We assign Gamma priors to $\boldsymbol{\lambda}$: $p(\boldsymbol{\lambda}) = \prod_{r=1}^R \text{Gamma}(\lambda_r|a_r^0, b_r^0)$, identical to (2). Then, the rank-revealing prior over all factor trajectories is:

$$p(\mathcal{U}, \boldsymbol{\lambda}) = p(\boldsymbol{\lambda}) \prod_{k=1}^K p(\mathbf{u}^k(i_k, t)|\boldsymbol{\lambda}), \tag{11}$$

where $\mathcal{U}$ denotes the set of factor trajectories $\{\mathbf{u}^k(\cdot, \cdot)\}_{k=1}^K$. *We are to emphasize that the sparsity-inducing prior is assigned over a family of latent functions, rather than over a set of static factors [8, 9]*. Consequently, classical inference techniques developed are not applicable in the functional tensor context. In the next section, we introduce a novel gradientdescentbased inference method that efficiently prunes redundant components from these latent functions.

With finite observed entries $\mathcal{D} = \{y_n, \mathbf{i}_n, t_n\}_{n=1}^N$, where $y_n$ denotes the $n$-$th$ entry observed at continuous index tuple $\mathbf{i}_n = (i_1^n, \cdots, i_K^n)$ and timestamp $t_n$. Our goal is to learn a factorized function as described in (5) to construct a direct mapping from $(\mathbf{i}_n, t_n)$ to $y_n$. Therefore, for each observed entry $\{y_n, \mathbf{i}_n, t_n\}$, we model the Gaussian likelihood as:

$$p(y_n|\mathcal{U}, \tau) = \mathcal{N}(y_n|\mathbf{1}^T[\underset{k}{\circledast} \mathbf{u}^k(i_k^n, t_n)], \tau^{-1}), \tag{12}$$

where $\tau$ is the inverse of the observation noise. We further assign a Gamma prior, $p(\tau) = \text{Gamma}(\tau|c^0, d^0)$, and the joint probability can be written as:

$$p(\mathcal{U}, \tau, \mathcal{D}) = p(\mathcal{U}, \boldsymbol{\lambda})p(\tau) \prod_{n=1}^N p(y_n|\mathcal{U}, \tau). \tag{13}$$

We illustrate CATTE with the case of $K = 3$ in Figure 1.

# 4 Model Inference

## 4.1 Factorized Functional Posterior and Analytical Evidence Lower Bound

It is intractable to compute the full posterior of latent variables in (13) due to the high-dimensional integral and complex form of likelihood. We take a workaround to construct a variational distribution $q(\mathcal{U}, \boldsymbol{\lambda}, \tau)$ to approximate the exact posterior $p(\mathcal{U}, \boldsymbol{\lambda}, \tau | \mathcal{D})$. Similar to the widely-used mean-field assumption, we design the approximate posterior in a fully factorized form: $q(\mathcal{U}, \boldsymbol{\lambda}, \tau) = q(\mathcal{U}) q(\boldsymbol{\lambda}) q(\tau)$.

Specifically, the conditional conjugate property of Gaussian-Gamma distribution motivates us to formulate the corresponding variational posteriors as follows:

$$q(\mathcal{U}) = \prod_{n=1}^{N} \prod_{k=1}^{K} \mathcal{N}(\mathbf{u}^k(i_k^n, t_n) | \mathbf{g}^k(i_k^n, t_n), \sigma^2 \mathbf{I}), \tag{14}$$

where $\mathbf{g}^k(\cdot, \cdot)$ is the mode-wise latent temporal representations parameterized by $\boldsymbol{\omega}_k$, as we mentioned in Section 3.1, and $\sigma$ is the variational variance shared by all $\mathbf{u}^k$. Similarly, we formulate $q(\boldsymbol{\lambda}), q(\tau)$ as:

$$q(\boldsymbol{\lambda}) = \prod_{r=1}^{R} \text{Gamma}(\lambda_r | \alpha_r, \beta_r), q(\tau) = \text{Gamma}(\tau | \rho, \iota), \tag{15}$$

where $\{\alpha_r, \beta_r\}_{r=1}^{R}, \rho, \iota$ are the variational parameters to characterize the approximated posteriors. Our goal is to estimate the latent ODE parameters $\boldsymbol{\omega}_k$ and variational parameters $\{\{\alpha_r, \beta_r\}_{r=1}^{R}, \sigma, \rho, \iota\}$ in (14)(15) to make the approximated posterior $q(\mathcal{U}, \boldsymbol{\lambda}, \tau)$ as close as possible to the true posterior $p(\mathcal{U}, \boldsymbol{\lambda}, \tau | \mathcal{D})$. To do so, we follow the variational inference framework [22] and construct the following objective function by minimizing the Kullback-Leibler (KL) divergence between the approximated posterior and the true posterior $\text{KL}(q(\mathcal{U}, \boldsymbol{\lambda}, \tau) \| p(\mathcal{U}, \boldsymbol{\lambda}, \tau | \mathcal{D}))$, which leads to the maximization of the evidence lower bound (ELBO):

$$\text{ELBO} = \mathbb{E}_{q(\mathcal{U}, \boldsymbol{\lambda}, \tau)}[\ln p(\mathcal{D} | \mathcal{U}, \boldsymbol{\lambda}, \tau)] + \mathbb{E}_{q(\mathcal{U}, \boldsymbol{\lambda})}[\ln \frac{p(\mathcal{U} | \boldsymbol{\lambda})}{q(\mathcal{U})}] - \text{KL}(q(\boldsymbol{\lambda}) \| p(\boldsymbol{\lambda})) - \text{KL}(q(\tau) \| p(\tau)). \tag{16}$$

The ELBO is consist of four terms. The first term is posterior expectation of log-likelihood while the last three are KL terms. Usually, the first term is intractable if the likelihood model is complicated and requires the costly sampling-based approximation to handle the integrals in the expectation [23, 15]. Fortunately, by leveraging the well-designed conjugate priors and factorized structure of the posterior, we make an endeavor to derive its analytical expression:

$$\mathbb{E}_{q(\mathcal{U}, \boldsymbol{\lambda}, \tau)}[\ln p(\mathcal{D} | \mathcal{U}, \boldsymbol{\lambda}, \tau)] = -\frac{N}{2} \ln(2\pi) + \frac{N}{2}(\psi(\rho) - \ln \iota)$$

$$-\frac{1}{2} \sum_{n=1}^{N} \frac{\rho}{\iota} \{ y_n^2 - 2y_n \{ \mathbf{1}^{\text{T}}[\underset{k}{\circledast} \mathbf{g}^k(i_k^n, t_n)] \} + \mathbf{1}^{\text{T}}[\underset{k}{\circledast} \text{vec}(\mathbf{g}^k(i_k^n, t_n) \mathbf{g}^k(i_k^n, t_n)^{\text{T}} + \sigma^2 \mathbf{I})] \}, \tag{17}$$

where $\mathbf{g}_r^k(i_k^n, t_n)$ is the $r$-th element of the $k$-th mode's latent temporal representation $\mathbf{g}^k(i_k^n, t_n)$. We provide the detailed derivation in Appendix A.2. The second term of (16) computes the KL divergence between prior and posterior of factor trajectories, which is also with a closed from:

$$\mathbb{E}_{q(\mathcal{U}, \boldsymbol{\lambda})}[\ln \frac{p(\mathcal{U} | \boldsymbol{\lambda})}{q(\mathcal{U})}] = -\text{KL}(q(\mathcal{U}) \| p(\mathcal{U} | \boldsymbol{\lambda} = \mathbb{E}_q(\boldsymbol{\lambda}))) =$$

$$-\sum_{n=1}^{N} \sum_{k=1}^{K} \sum_{r=1}^{R} \frac{1}{2} \{ \ln(\frac{\beta_r}{\alpha_r \sigma^2}) + \frac{\alpha_r}{\beta_r} \{ \sigma^2 + [\mathbf{g}_r^k(i_k^n, t_n)]^2 \} - 1 \}, \tag{18}$$

where $\mathbb{E}_q(\boldsymbol{\lambda}) = [\mathbb{E}_q(\lambda_1), \ldots, \mathbb{E}_q(\lambda_R)]^{\text{T}} = [\alpha_1/\beta_1, \ldots, \alpha_R/\beta_R]^{\text{T}}$. This term encourages rank reduction, as it drives the posterior mean of $\lambda_r$ to be large, thereby forcing the corresponding $r$-th component of $K$ factor trajectories $\{\mathbf{g}_r^k(\cdot, \cdot)\}_{k=1}^{K}$ to be zero. The combination of the above two terms enables an automatic rank determination mechanism by striking a balance between capacity of representation and model complexity. As the prior and posterior of $\boldsymbol{\lambda}$ and $\tau$ are both Gamma distribution, the last two KL terms in the ELBO are analytically computable, as shown in (26)(27) in Appendix A.2.

*We highlight that each term in* (16) *admits a closedform expression, eliminating the need for samplingbased approximations.* Consequently, during training we can compute the ELBOs gradient with respect to the variational parameters exactly, allowing us to employ standard gradientbased optimization methods. We present the differences from previous methods in Appendix C. Our proposed method scales efficiently to jointly optimize both the variational and latent ODE parameters:

$$\text{argmax}_{\{\boldsymbol{\omega}_k\}_{k=1}^K, \{\alpha_r, \beta_r\}_{r=1}^R, \sigma, \rho, \iota} \text{ELBO.} \tag{19}$$

We summarize the inference algorithm in Algorithm 1. When $N$ is large, we can use the minibatch gradient descent method to accelerate the optimization process. After obtaining the variational posteriors of the latent variables, we can further derive the predictive distribution for new data at arbitrary indices. More details can be found in Appendix A.6. To distinguish our method from previous approaches, we refer to it as a functional automatic rank determination mechanism (FARD).

## 5 Related Work

Many existing approaches augment the tensor with a time mode to integrate temporal information [24, 25, 26, 27]. To leverage continuous timestamps, [4] modeled the tensor core while [5] modeled the latent factors as time functions with Gaussian processes within the Tucker decomposition. Similarly, [15] captured the temporal evolution of latent factors in the frequency domain by employing a Gaussian process prior, with the factor trajectories subsequently generated via inverse Fourier transform. In contrast, [6] directly applied a neural ODE model to represent tensor entry values as a function of the corresponding latent factors and time. However, it learns time-invariant latent factors to control the derivatives of the entry dynamics, limiting its expressiveness. The most recent work [28] constructed a multipartite graph to encode interactions across different modes into the evolution of latent factors. However, the above methods are unable to model the temporal tensor data with all modes being continuous-indexed.

Within burgeoning literature on functional tensors, existing methods often rely on deterministic tensor models, such as the Tucker model [17] or tensor train models [29, 30, 18, 31, 32], to approximate multivariate functions with low-rank structures. However, these methods are sensitive to data noise and lack the capability to provide uncertainty quantification. Alternatively, tensor-based Bayesian models with Gaussian process priors have been proposed to represent continuous-indexed latent factors [33, 19]. Despite their advantages, these approaches do not explicitly model temporal dynamics, limiting their effectiveness in capturing complex patterns.

## 6 Experiment

### 6.1 Synthetic Data

We first evaluated CATTE on a synthetic task. We generated a two-mode temporal tensor, and each entry is defined as:

$$\boldsymbol{\mathcal{Y}}(i_1, i_2, t) = \mathbf{1}^{\text{T}}[\mathbf{u}^1(i_1, t) \circledast \mathbf{u}^2(i_2, t)],$$
$$\text{where} \quad \mathbf{u}^1(i_1, t) = -\cos^3(2\pi t + 2.5\pi i_1); \quad \mathbf{u}^2(i_2, t) = \sin(3\pi t + 3.5\pi i_2). \tag{20}$$

We randomly sampled $25 \times 25 \times 50$ off-grid indexes entries from interval $[0, 1] \times [0, 1] \times [0, 1]$. We added Gaussian noise $\epsilon \sim \mathcal{N}(0, 0.05)$ to the generated data. We randomly selected $20\%$ of the data (6250 points in total) as the training data. Detailed model settings can be found in Appendix B.1. In Figure 2, we showed the predictive trajectories of entry value indexed in different coordinates. The dotted line represents the ground truth and the full line represents the the predictive mean learned by our model. The cross symbols represent the training points. The shaded region represents the predictive uncertainty region.

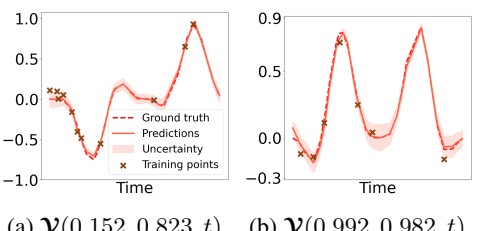

(a) $\boldsymbol{\mathcal{Y}}(0.152, 0.823, t)$ (b) $\boldsymbol{\mathcal{Y}}(0.992, 0.982, t)$

Figure 2: Prediction results on different coordinates.

One can see that although the training points are sparse and noisy, CATTE accurately recovered the ground truth, demonstrating that it has effectively captured the temporal dynamics. Figure 3 depicts

$R$ components of the learned factor trajectories at timestamp $t = 0.235$. One can see that CATTE identifies the underlying rank (i.e., 1) through uniquely recovering the real mode functions and other four components are learned to be zero. More detailed interpretations on the rank revealing process were provided in Appendix B.3.

## 6.2 Real-world Data

**Datasets:** We examined CATTE on three real-world benchmark datasets. (1) CA traffic, lane-blocked records in California. We extracted a three-mode temporal tensor between 5 severity levels, 20 latitudes and 16 longitudes. We collected 10K entry values and their timestamps.(`https://smoosavi.org/datasets/lstw`; (2) Server Room, temperature logs of Poznan Supercomputing and Networking Center. We extracted a three-mode temporal tensor between 3 air conditioning modes ($24°$, $27°$ and $30°$), 3 power usage levels ($50\%, 75\%, 100\%$) and 34 locations. We collected 10K entry values and their timestamps.(`https://zenodo.org/record/3610078#%23.Y8SYt3bMJGi`); (3) SSF, sound speed field measurements in the pacific ocean covering the region between latitudes $17°N \sim 20°N$, longitude $114.7°E \sim 117.7°E$ and depth $0m \sim 200m$. We extracted a three-mode continuous-indexed temporal tensor data contains 10K observations across 10 latitudes, 20 longitudes, 10 depths and 34 timestamps over 4 days. (`https://ncss.hycom.org/thredds/ncss/grid/GLBy0.08/expt_93.0/ts3z/dataset.html`).

**Baselines and Settings:** We compared CATTE with state-of-the-art temporal and functional tensor methods: (1) THIS-ODE [6], a continuous-time decomposition using a neural ODE to estimate tensor entries from static factors and time; (2) NONFAT [15], a bi-level latent GP model that estimates dynamic factors with Fourier bases; (3) DEMOTE [28], a neural diffusion-reaction process model for learning dynamic factors in tensor decomposition; (4) FunBaT [19], a Bayesian method using GPs as functional priors for continuous-indexed tensor data; (5) LRTFR [17], a low-rank functional Tucker model that uses factorized neural representations for decomposition. We followed [28, 19] to randomly

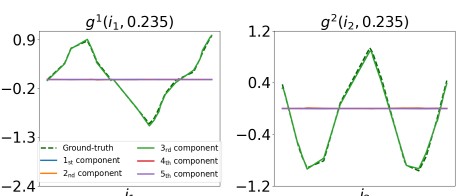

Figure 3: Visualizations of learned factor trajectories at timestamp $t = 0.235$. Only the $3_{rd}$ component is revealed to be informative and others are pruned to be zero.

draw $80\%$ of observed entries for training and the rest for testing. The performance metrics include the root-mean-square error (RMSE) and the mean absolute error (MAE). Each experiment was conducted five times (5fold crossvalidation) and we reported the average test errors with their standard deviations. For CATTE , we set the ODE state dimension $J = 10$ and the initial number of components of the factor trajectories $R = 10$. We provided more detailed baseline settings in Appendix B.2.

**Prediction Performance:** Table 1 shows that CATTE consistently outperforms all baselines by a substantial margin, without requiring manual rank tuning. Moreover, the integration of FARD leads to significant performance gains, highlighting its effectiveness. The learned ranks of the CA traffic, Server Room and SSF datasets are $5, 7, 7$ respectively. We illustrated their rank-learning curves of three datasets in Figure 6 in Appendix B.3. We observed that methods which do not consider the continuously indexed mode (e.g., NONFAT, DEMOTE, THIS-ODE) perform poorly on the SSF dataset. In contrast, approaches that leverage this continuity achieve significantly better results. This is because the SSF dataset exhibits strong continuity across three modes, and methods that fail to incorporate this information struggle to deliver satisfactory reconstructions. Additional visualization results of the predictions from various methods and datasets are presented in Fig. 7, Fig. 8 and Fig. 9, and in Appendix B.7.

**Revealed Rank Analysis and Interpretability:** We analyzed the revealed rank and the learned factor trajectories of the SSF dataset. Figure 4(a) shows the posterior mean of the variance of the learned factor trajectories (i.e., $\mathbb{E}_q(\frac{1}{\lambda})$), which governs the fluctuations of their corresponding $R = 10$ components of factor trajectories. One can see that $\mathbb{E}_q(\frac{1}{\lambda_3}), \mathbb{E}_q(\frac{1}{\lambda_4})$ and $\mathbb{E}_q(\frac{1}{\lambda_8})$ are small, indicating that these components concentrate around zero and can be pruned without affecting the final predictions. Thus, our method revealed the rank of the SSF dataset to be 7. Additionally, $\mathbb{E}_q(\frac{1}{\lambda_1})$ and $\mathbb{E}_q(\frac{1}{\lambda_{10}})$ dominate, indicating that the corresponding $1_{st}$ and $10_{th}$ components of the factor trajectories form

| Datasets | RMSE | | | MAE | | |
|---|---|---|---|---|---|---|
| | *CA Traffic* | *Server Room* | *SSF* | *CA Traffic* | *Server Room* | *SSF* |
| | | | $R = 3$ | | | |
| THIS-ODE | $0.672 \pm 0.002$ | $0.132 \pm 0.002$ | $2.097 \pm 0.003$ | $0.587 \pm 0.002$ | $0.083 \pm 0.002$ | $2.084 \pm 0.003$ |
| NONFAT | $0.504 \pm 0.010$ | $0.129 \pm 0.002$ | $9.796 \pm 0.010$ | $0.167 \pm 0.009$ | $0.078 \pm 0.001$ | $8.771 \pm 0.043$ |
| DEMOTE | $0.447 \pm 0.001$ | $0.131 \pm 0.001$ | $9.789 \pm 0.001$ | $0.118 \pm 0.002$ | $0.090 \pm 0.0015$ | $8.757 \pm 0.001$ |
| FunBaT-CP | $0.563 \pm 0.025$ | $0.425 \pm 0.003$ | $0.696 \pm 0.047$ | $0.244 \pm 0.025$ | $0.308 \pm 0.001$ | $0.549 \pm 0.038$ |
| FunBaT-Tucker | $0.584 \pm 0.009$ | $0.498 \pm 0.058$ | $0.730 \pm 0.201$ | $0.189 \pm 0.014$ | $0.381 \pm 0.053$ | $0.614 \pm 0.128$ |
| LRTFR | $0.379 \pm 0.042$ | $0.151 \pm 0.004$ | $0.595 \pm 0.018$ | $0.187 \pm 0.022$ | $0.110 \pm 0.002$ | $0.464 \pm 0.0165$ |
| | | | $R = 5$ | | | |
| THIS-ODE | $0.632 \pm 0.002$ | $0.132 \pm 0.003$ | $1.039 \pm 0.015$ | $0.552 \pm 0.001$ | $0.083 \pm 0.002$ | $1.032 \pm 0.002$ |
| NONFAT | $0.501 \pm 0.002$ | $0.117 \pm 0.006$ | $9.801 \pm 0.014$ | $0.152 \pm 0.001$ | $0.071 \pm 0.004$ | $8.744 \pm 0.035$ |
| DEMOTE | $0.421 \pm 0.002$ | $0.105 \pm 0.003$ | $9.788 \pm 0.001$ | $0.103 \pm 0.001$ | $0.068 \pm 0.003$ | $8.757 \pm 0.001$ |
| FunBaT-CP | $0.547 \pm 0.025$ | $0.422 \pm 0.001$ | $0.675 \pm 0.061$ | $0.204 \pm 0.052$ | $0.307 \pm 0.002$ | $0.531 \pm 0.051$ |
| FunBaT-Tucker | $0.578 \pm 0.005$ | $0.521 \pm 0.114$ | $0.702 \pm 0.054$ | $0.181 \pm 0.005$ | $0.391 \pm 0.097$ | $0.557 \pm 0.041$ |
| LRTFR | $0.376 \pm 0.016$ | $0.167 \pm 0.006$ | $0.532 \pm 0.036$ | $0.182 \pm 0.012$ | $0.121 \pm 0.005$ | $0.418 \pm 0.003$ |
| | | | $R = 7$ | | | |
| THIS-ODE | $0.628 \pm 0.007$ | $0.154 \pm 0.016$ | $1.685 \pm 0.009$ | $0.548 \pm 0.006$ | $0.089 \pm 0.002$ | $1.674 \pm 0.008$ |
| NONFAT | $0.421 \pm 0.016$ | $0.128 \pm 0.002$ | $9.773 \pm 0.015$ | $0.137 \pm 0.006$ | $0.077 \pm 0.002$ | $8.718 \pm 0.035$ |
| DEMOTE | $0.389 \pm 0.005$ | $0.094 \pm 0.006$ | $9.790 \pm 0.002$ | $0.091 \pm 0.001$ | $0.062 \pm 0.006$ | $8.753 \pm 0.006$ |
| FunBaT-CP | $0.545 \pm 0.009$ | $0.426 \pm 0.001$ | $0.685 \pm 0.049$ | $0.204 \pm 0.037$ | $0.307 \pm 0.001$ | $0.541 \pm 0.039$ |
| FunBaT-Tucker | $0.587 \pm 0.011$ | $0.450 \pm 0.041$ | $0.642 \pm 0.037$ | $0.195 \pm 0.022$ | $0.330 \pm 0.026$ | $0.507 \pm 0.029$ |
| LRTFR | $0.365 \pm 0.042$ | $0.156 \pm 0.012$ | $0.502 \pm 0.033$ | $0.161 \pm 0.014$ | $0.118 \pm 0.009$ | $0.392 \pm 0.028$ |
| | | | Functional Automatic Rank Determination | | | |
| CATTE (Ours) | $\mathbf{0.284 \pm 0.016}$ | $\mathbf{0.078 \pm 0.001}$ | $\mathbf{0.373 \pm 0.003}$ | $\mathbf{0.085 \pm 0.004}$ | $\mathbf{0.047 \pm 0.003}$ | $\mathbf{0.288 \pm 0.003}$ |
| CATTE w.o. FARD | $0.301 \pm 0.020$ | $0.091 \pm 0.008$ | $0.402 \pm 0.013$ | $0.094 \pm 0.010$ | $0.0657 \pm 0.005$ | $0.310 \pm 0.010$ |

Table 1: Predictive errors and standard deviation. The results were averaged over five runs.

(a) Posterior mean of $\frac{1}{\lambda}$    (b) Factor trajectories    (c) Entry predictions    (d) Scalability

Figure 4: Illustrations on (a) the posterior mean of the variance from the SSF dataset ($R = 10$), (b) the dominant components of learned depth-mode factor trajectories from the SSF dataset, (c) the entry value predictions indexed in ($17°$N, $114.7°$E, 30m) of the SSF dataset, (d) the scalability over the length of time series on synthetic dataset.

the primary structure of the data. To illustrate this, we plotted these two components of the depth-mode factor trajectories at depth 30m in Figure 4(b). As we can see, the trajectories show periodic patterns, which are influenced by the the day-night cycle of ocean temperature. We also compared the predicted curves of CATTE and LRTFR for an entry located at $17°$N, $114.7°$E and a depth of 30m. CATTE outperforms LRTFR, providing more accurate predictions with uncertainty quantification, even outside the training region (right to the dashed vertical line). This suggests that our model holds promise for extrapolation tasks. The results collectively highlighted the advantage of CATTE in capturing continuous-indexed multidimensional dynamics, which is crucial for analyzing real-world temporal data and performing predictive tasks.

| | CATTE | CATTE w.o. FARD | LRTFR (R=5) | DEMOTE (R=5) |
|---|---|---|---|---|
| Noise Variance | RMSE | | | |
| 0.5 | $\mathbf{0.305 \pm 0.005}$ | $0.356 \pm 0.010$ | $0.434 \pm 0.047$ | $0.430 \pm 0.010$ |
| 1 | $\mathbf{0.406 \pm 0.007}$ | $0.461 \pm 0.015$ | $0.516 \pm 0.034$ | $0.552 \pm 0.009$ |

Table 2: Experiments on the robustness of functional automatic rank determination mechanism against the noise on the CA traffic dataset. The results were averaged over five runs.

**Robustness against Noise:** The incorporated FARD mechanism can reveal the underlying rank of the functional temporal data and prune the unnecessary components of the factor trajectories, enhancing noise robustness. To evaluate its performance, we added Gaussian noise with varying variance levels to the training set of CA traffic dataset and compared the results of different methods, as summarized in Table 2. We disabled FARD by using a simple RMSE criterion as the objective function to constitute an ablation study. More detailed results of the baselines with varying ranks

in Table 4 and Table 5 in Appendix B.4. CATTE achieves lower prediction errors than CATTE w.o. FARD, , confirming FARDs benefit. We also demonstrate the robustness of CATTE aganist Laplacian and Poisson noise in Table 6 (Appendix B.4).

| Tensor size | $20 \times 20 \times 20 \times 25$ (4th-oder) | $20 \times 20 \times 20 \times 20 \times 25$ (5th-oder) | $20 \times 20 \times 20 \times 20 \times 20 \times 25$ (6th-oder) |
|---|---|---|---|
| Time(s) | 0.433 | 1.101 | 3.062 |

Table 3: Per-epoch/iteration running time on different orders of tensors(in seconds).

**Sensitivity and Scalability:** We examined the sensitivity of CATTE with respect to the variational hyperparamter $a_r^0$, $b_r^0$ and the dimensionality of ODE state $J$ on the CA traffic dataset. The results were given in Table 9 and Table 8 respectively. Empirically, CATTE performs consistently well across different $a_r^0$, $b_r^0$ and $J$. We further evaluated the scalability of CATTE with respect to the length of the time series $T$ and $J$. For $T$, we randomly sampled $25 \times 25 \times T$ off-grid index entries from the interval $[0, 1] \times [0, 1] \times [0, 1]$ using Eq. (20) for training. We varied $T$ across $\{50, 200, 500, 1000\}$ and $J$ across $\{5, 25, 30\}$. To better quantify the scalability, we employed the simple Euler scheme with fixed step size [34] for network training. The results shown in Figure 4(d) indicate that the running time of CATTE grows linearly with $T$ and is insensitive to $J$, demonstrating its suitability for large-scale applications.

In addition, we conducted experiments to assess scalability with respect to tensor order. We generated synthetic datasets with varying orders while fixing the temporal dimension to 25 and the non-temporal dimensions to 20. From each dataset, we randomly extracted $1\%$ of the total data points for training. The average training time per epoch is reported in Table 3. The results show that CATTE scales well with tensor order. By decoupling each mode, CATTE translates the addition of modes into an increase in the number of ODE states under our proposed efficiency-driven approach (as discussed in Section 3.1). For example, for a 4th-order tensor of size $20 \times 20 \times 20 \times 25$, we only need to solve $20 + 20 + 20 = 60$ ODE states, which can be updated simultaneously with a single ODE solver. Additional results on the scalability of CATTE are provided in Appendix B.8.

**Computational Efficiency:** We compared per-iteration runtimes of CATTE and baselines on a system with an NVIDIA RTX 4070 GPU, Intel i9-13900H CPU, 32 GB RAM, and 1 TB SSD (Table 7, Appendix B.5). CATTE is slightly faster than NONFAT and DEMOTE, and much faster than THIS-ODE.

## 7 Conclusion

We have presented CATTE, a complexity-adaptive model for functional temporal tensor decomposition. CATTE automatically infers the underlying rank of temporal tensors with continuously indexed modes, while achieving state-of-the-art predictive performance and capturing interpretable temporal patterns efficiently. A current limitation of our work is the lack of explicit incorporation of physical laws into the modeling process. In future work, we plan to integrate CATTE with domain-specific physical knowledge, enabling more accurate long-range and wide-area physical field reconstructions.

## 8 Acknowledgment

The work was supported in part by the National Natural Science Foundation of China under Grant 62371418, in part by the Fundamental Research Funds for the Central Universities (226-2025-00168) and in part by Zhejiang University Education Foundation Qizhen Scholar Foundation.

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

# APPEXNDIX

## A  Details of derivations

### A.1  Log-marginal likelihood

$$
\begin{aligned}
\ln p(\mathcal{D}) &= \int q(\mathcal{U}, \boldsymbol{\lambda}, \tau) \ln p(\mathcal{D}) d\mathcal{U} d\boldsymbol{\lambda} d\tau = \int q(\mathcal{U}, \boldsymbol{\lambda}, \tau) \ln \frac{p(\mathcal{U}, \boldsymbol{\lambda}, \tau, \mathcal{D})}{p(\mathcal{U}, \boldsymbol{\lambda}, \tau | \mathcal{D})} d\mathcal{U} d\boldsymbol{\lambda} d\tau \\
&= \int q(\mathcal{U}, \boldsymbol{\lambda}, \tau) \ln \frac{p(\mathcal{U}, \boldsymbol{\lambda}, \tau, \mathcal{D}) q(\mathcal{U}, \boldsymbol{\lambda}, \tau)}{p(\mathcal{U}, \boldsymbol{\lambda}, \tau | \mathcal{D}) q(\mathcal{U}, \boldsymbol{\lambda}, \tau)} d\mathcal{U} d\boldsymbol{\lambda} d\tau \\
&= \int q(\mathcal{U}, \boldsymbol{\lambda}, \tau) \ln \frac{p(\mathcal{U}, \boldsymbol{\lambda}, \tau, \mathcal{D})}{q(\mathcal{U}, \boldsymbol{\lambda}, \tau)} d\mathcal{U} d\boldsymbol{\lambda} d\tau - \int q(\mathcal{U}, \boldsymbol{\lambda}, \tau) \ln \frac{p(\mathcal{U}, \boldsymbol{\lambda}, \tau | \mathcal{D})}{q(\mathcal{U}, \boldsymbol{\lambda}, \tau)} d\mathcal{U} d\boldsymbol{\lambda} d\tau \quad (21) \\
&= \mathbb{E}_{q(\mathcal{U}, \boldsymbol{\lambda}, \tau)}[\ln \frac{p(\mathcal{U}, \boldsymbol{\lambda}, \tau, \mathcal{D})}{q(\mathcal{U}, \boldsymbol{\lambda}, \tau)}] - \mathbb{E}_{q(\mathcal{U}, \boldsymbol{\lambda}, \tau)}[\ln \frac{p(\mathcal{U}, \boldsymbol{\lambda}, \tau) | \mathcal{D}}{q(\mathcal{U}, \boldsymbol{\lambda}, \tau)}] \\
&= \mathcal{L}(q) + \mathrm{KL}(q(\mathcal{U}, \boldsymbol{\lambda}, \tau) \| p(\mathcal{U}, \boldsymbol{\lambda}, \tau | \mathcal{D})).
\end{aligned}
$$

### A.2  Lower bound of log-marginal likelihood

$$
\begin{aligned}
\mathcal{L}(q) &= \mathbb{E}_{q(\mathcal{U}, \boldsymbol{\lambda}, \tau)}[\ln \frac{p(\mathcal{U}, \boldsymbol{\lambda}, \tau, \mathcal{D})}{q(\mathcal{U}, \boldsymbol{\lambda}, \tau)}] = \mathbb{E}_{q(\mathcal{U}, \boldsymbol{\lambda}, \tau)}[\ln p(\mathcal{U}, \boldsymbol{\lambda}, \tau, \mathcal{D})] - \mathbb{E}_{q(\mathcal{U}, \boldsymbol{\lambda}, \tau)}[\ln q(\mathcal{U}, \boldsymbol{\lambda}, \tau)] \\
&= \mathbb{E}_{q(\mathcal{U}, \boldsymbol{\lambda}, \tau)}[\ln p(\mathcal{D} | \mathcal{U}, \boldsymbol{\lambda}, \tau)] + \mathbb{E}_{q(\mathcal{U}, \boldsymbol{\lambda}, \tau)}[\ln p(\mathcal{U}, \boldsymbol{\lambda}, \tau)] - \mathbb{E}_{q(\mathcal{U}, \boldsymbol{\lambda}, \tau)}[\ln q(\mathcal{U}, \boldsymbol{\lambda}, \tau)] \\
&= \mathbb{E}_{q(\mathcal{U}, \boldsymbol{\lambda}, \tau)}[\ln p(\mathcal{D} | \mathcal{U}, \boldsymbol{\lambda}, \tau)] + \mathbb{E}_{q(\mathcal{U}, \boldsymbol{\lambda}, \tau)}[\ln \frac{p(\mathcal{U} | \boldsymbol{\lambda})}{q(\mathcal{U})} + \ln \frac{p(\boldsymbol{\lambda})}{q(\boldsymbol{\lambda})} + \ln \frac{p(\tau)}{q(\tau)}] \\
&= \mathbb{E}_{q(\mathcal{U}, \boldsymbol{\lambda}, \tau)}[\ln p(\mathcal{D} | \mathcal{U}, \boldsymbol{\lambda}, \tau)] + \mathbb{E}_{q(\mathcal{U}, \boldsymbol{\lambda})}[\ln \frac{p(\mathcal{U} | \boldsymbol{\lambda})}{q(\mathcal{U})}] - \mathrm{KL}(q(\boldsymbol{\lambda}) \| p(\boldsymbol{\lambda})) - \mathrm{KL}(q(\tau) \| p(\tau)).
\end{aligned}
\quad (22)
$$

The first term of evidence lower bound (posterior expectation of log-likelihood) can be written as:

$$
\begin{aligned}
\mathbb{E}_{q(\mathcal{U}, \boldsymbol{\lambda}, \tau)}[\ln p(\mathcal{D} | \mathcal{U}, \boldsymbol{\lambda}, \tau)] &= -\frac{N}{2} \ln(2\pi) + \frac{N}{2} \mathbb{E}_q[\ln \tau] - \frac{1}{2} \sum_{n=1}^{N} \mathbb{E}_q[\tau] \mathbb{E}_q[(y_n - \mathbf{1}^{\mathrm{T}} \underset{k}{\circledast} \mathbf{u}^k(i_k^n, t_n))^2] \\
&= -\frac{N}{2} \ln(2\pi) + \frac{N}{2} (\psi(\rho) - \ln \iota) - \frac{1}{2} \sum_{n=1}^{N} \frac{\rho}{\iota} \mathbb{E}_q[(y_n - \mathbf{1}^{\mathrm{T}} \underset{k}{\circledast} \mathbf{u}^k(i_k^n, t_n))^2],
\end{aligned}
\quad (23)
$$

where $\psi(\cdot)$ is the digamma function. The posterior expectation of model error is:

$$
\mathbb{E}_q[(y_n - \mathbf{1}^{\mathrm{T}} \underset{k}{\circledast} \mathbf{u}^k(i_k^n, t_n))^2] = y_n^2 - 2 y_n \mathbf{1}^{\mathrm{T}} \underset{k}{\circledast} \mathbf{g}^k(i_k^n, t_n) + \mathbf{1}^{\mathrm{T}} \underset{k}{\circledast} \mathrm{vec}(\mathbf{g}^k(i_k^n, t_n) \mathbf{g}^k(i_k^n, t_n)^{\mathrm{T}} + \sigma^2 \mathbf{I}).
\quad (24)
$$

If $\sigma = 0$, then posterior expectation of model error becomes $(y_n - \mathbf{1}^{\mathrm{T}} \underset{k}{\circledast} \mathbf{u}^k(i_k^n, t_n))^2$.

The second term of evidence lower bound can be written as[2]:

$$
\begin{aligned}
\mathbb{E}_{q(\mathcal{U}, \boldsymbol{\lambda})}[\ln \frac{p(\mathcal{U} | \boldsymbol{\lambda})}{q(\mathcal{U})}] &= \int \int q(\mathcal{U}, \boldsymbol{\lambda}) \ln \frac{p(\mathcal{U} | \boldsymbol{\lambda})}{q(\mathcal{U})} d\mathcal{U} d\boldsymbol{\lambda} = \int \int q(\mathcal{U}, \boldsymbol{\lambda}) \ln p(\mathcal{U} | \boldsymbol{\lambda}) d\mathcal{U} d\boldsymbol{\lambda} - \int q(\mathcal{U}) \ln q(\mathcal{U}) d\mathcal{U} \\
&= \int q(\mathcal{U}) \ln p(\mathcal{U} | \boldsymbol{\lambda} = \mathbb{E}_q(\boldsymbol{\lambda})) d\mathcal{U} - \int q(\mathcal{U}) \ln q(\mathcal{U}) d\mathcal{U} \\
&= -\mathrm{KL}(q(\mathcal{U}) \| p(\mathcal{U} | \boldsymbol{\lambda} = \mathbb{E}_q(\boldsymbol{\lambda}))) = -\sum_{n=1}^{N} \sum_{k=1}^{K} \sum_{r=1}^{R} \frac{1}{2} [\ln(\frac{\beta_r}{\alpha_r \sigma^2}) + \frac{\alpha_r}{\beta_r} (\sigma^2 + (\mathbf{g}_r^k(i_k^n, t_n))^2) - 1],
\end{aligned}
\quad (25)
$$

where $\mathbf{g}_r^k(i_k^n, t_n)$ is the $r$-th element of $\mathbf{g}^k(i_k^n, t_n)$.

---

[2]The KL divergence between two Gaussian distributions $p(x) \sim \mathcal{N}(x | \mu_1, \sigma_1^2)$ and $q(x) \sim \mathcal{N}(x | \mu_2, \sigma_2^2)$ can be computed using $\mathrm{KL}(p \| q) = \frac{1}{2} [\ln(\frac{\sigma_2^2}{\sigma_1^2}) + \frac{\sigma_1^2 + (\mu_1 - \mu_2)^2}{\sigma_2^2} - 1]$. Detailed derivation can be found in Appendix A.3.

The third term of evidence lower bound can be written as[3]:

$$\text{KL}(q(\boldsymbol{\lambda})\|p(\boldsymbol{\lambda})) = \sum_{r=1}^{R} a_0 \ln \frac{\beta_r}{b_0} - \ln \frac{\Gamma(a_0)}{\Gamma(\alpha_r)} + (\alpha_r - a_0)\psi(\alpha_r) - (b_0 - b_1)\frac{\alpha_r}{\beta_r}. \qquad (26)$$

The fourth term of evidence lower bound can be written as:

$$\text{KL}(q(\tau)\|p(\tau)) = c_0 \ln \frac{\iota}{d_0} - \ln \frac{\Gamma(c_0)}{\Gamma(\rho)} + (\rho - c_0)\psi(\rho) - (d_0 - \iota)\frac{\rho}{\iota}. \qquad (27)$$

### A.3 KL divergence of two Gaussian distribution

The Kullback-Leibler (KL) Divergence between two probability distributions $p$ and $q$ is defined as:

$$\text{KL}(p\|q) = \int_{-\infty}^{\infty} p(x) \ln\left(\frac{p(x)}{q(x)}\right) dx.$$

Let $p \sim \mathcal{N}(\mu_1, \sigma_1^2)$ and $q \sim \mathcal{N}(\mu_2, \sigma_2^2)$, where the probability density functions (PDFs) are given by:

$$p(x) = \frac{1}{\sqrt{2\pi\sigma_1^2}} \exp\left(-\frac{(x-\mu_1)^2}{2\sigma_1^2}\right),$$

$$q(x) = \frac{1}{\sqrt{2\pi\sigma_2^2}} \exp\left(-\frac{(x-\mu_2)^2}{2\sigma_2^2}\right).$$

Substitute the Gaussian PDFs into the definition of KL divergence:

$$\text{KL}(p\|q) = \int_{-\infty}^{\infty} \frac{1}{\sqrt{2\pi\sigma_1^2}} \exp\left(-\frac{(x-\mu_1)^2}{2\sigma_1^2}\right) \ln\left(\frac{\frac{1}{\sqrt{2\pi\sigma_1^2}} \exp\left(-\frac{(x-\mu_1)^2}{2\sigma_1^2}\right)}{\frac{1}{\sqrt{2\pi\sigma_2^2}} \exp\left(-\frac{(x-\mu_2)^2}{2\sigma_2^2}\right)}\right) dx.$$

Simplify the logarithmic term:

$$\ln\left(\frac{p(x)}{q(x)}\right) = \ln\left(\frac{\frac{1}{\sqrt{2\pi\sigma_1^2}}}{\frac{1}{\sqrt{2\pi\sigma_2^2}}}\right) + \left(-\frac{(x-\mu_1)^2}{2\sigma_1^2} + \frac{(x-\mu_2)^2}{2\sigma_2^2}\right)$$

$$= \ln\left(\frac{\sigma_2}{\sigma_1}\right) + \left(-\frac{(x-\mu_1)^2}{2\sigma_1^2} + \frac{(x-\mu_2)^2}{2\sigma_2^2}\right).$$

Thus, the integral for KL divergence becomes:

$$\text{KL}(p\|q) = \int_{-\infty}^{\infty} \frac{1}{\sqrt{2\pi\sigma_1^2}} \exp\left(-\frac{(x-\mu_1)^2}{2\sigma_1^2}\right) \left(\ln\left(\frac{\sigma_2}{\sigma_1}\right) + \left(-\frac{(x-\mu_1)^2}{2\sigma_1^2} + \frac{(x-\mu_2)^2}{2\sigma_2^2}\right)\right) dx.$$

Simplifying the Integral: We can now break the integral into two parts: 1. The constant term:

$$\int_{-\infty}^{\infty} \frac{1}{\sqrt{2\pi\sigma_1^2}} \exp\left(-\frac{(x-\mu_1)^2}{2\sigma_1^2}\right) \ln\left(\frac{\sigma_2}{\sigma_1}\right) dx.$$

Since $\frac{1}{\sqrt{2\pi\sigma_1^2}} \exp\left(-\frac{(x-\mu_1)^2}{2\sigma_1^2}\right)$ is the PDF of a Gaussian distribution, its integral is 1, so this term evaluates to:

$$\ln\left(\frac{\sigma_2}{\sigma_1}\right).$$

2. The difference of squared terms:

$$\int_{-\infty}^{\infty} \frac{1}{\sqrt{2\pi\sigma_1^2}} \exp\left(-\frac{(x-\mu_1)^2}{2\sigma_1^2}\right) \left(-\frac{(x-\mu_1)^2}{2\sigma_1^2} + \frac{(x-\mu_2)^2}{2\sigma_2^2}\right) dx.$$

---

[3]The KL divergence between two Gamma distributions $p(x) \sim \text{Gamma}(x|a_1, b_1)$ and $q(x) \sim \text{Gamma}(x|a_2, b_2)$ can be computed using $\text{KL}(p \| q) = a_2 \ln \frac{b_1}{b_2} - \ln \frac{\Gamma(a_2)}{\Gamma(a_1)} + (a_1 - a_2)\psi(a_1) - (b_2 - b_1)\frac{a_1}{b_1}$. Detailed derivation can be found in A.4.

This term can be split into two parts: - The first part involves $\mu_1$, and after calculation, it simplifies to:

$$\frac{\sigma_1^2}{2\sigma_2^2} - \frac{1}{2}.$$

The second part involves the difference between $\mu_1$ and $\mu_2$, and after calculation, it simplifies to:

$$\frac{(\mu_1 - \mu_2)^2}{2\sigma_2^2}.$$

Combining all parts, the KL divergence between two Gaussian distributions is:

$$\mathrm{KL}(p\|q) = \ln\left(\frac{\sigma_2}{\sigma_1}\right) + \frac{\sigma_1^2}{2\sigma_2^2} - \frac{1}{2} + \frac{(\mu_1 - \mu_2)^2}{2\sigma_2^2}.$$

### A.4 KL divergence of two Gamma distribution

The Kullback-Leibler (KL) Divergence between two probability distributions $p$ and $q$ is defined as:

$$\mathrm{KL}(p \parallel q) = \int_0^\infty p(x) \ln\left(\frac{p(x)}{q(x)}\right) dx.$$

Let $p \sim \mathrm{Gamma}(a_1, b_1)$ and $q \sim \mathrm{Gamma}(a_2, b_2)$, where the probability density functions (PDFs) with rate parameters are given by:

$$p(x) = \frac{b_1^{a_1} x^{a_1 - 1} \exp\left(-b_1 x\right)}{\Gamma(a_1)}, \quad x \geq 0,$$

$$q(x) = \frac{b_2^{a_2} x^{a_2 - 1} \exp\left(-b_2 x\right)}{\Gamma(a_2)}, \quad x \geq 0.$$

Substitute the PDFs of the Gamma distributions into the definition of KL divergence:

$$\mathrm{KL}(p \parallel q) = \int_0^\infty \frac{b_1^{a_1} x^{a_1 - 1} \exp\left(-b_1 x\right)}{\Gamma(a_1)} \ln\left(\frac{\frac{b_1^{a_1} x^{a_1 - 1} \exp(-b_1 x)}{\Gamma(a_1)}}{\frac{b_2^{a_2} x^{a_2 - 1} \exp(-b_2 x)}{\Gamma(a_2)}}\right) dx.$$

Simplify the logarithmic term:

$$\ln\left(\frac{p(x)}{q(x)}\right) = \ln\left(\frac{b_1^{a_1} x^{a_1 - 1} \exp\left(-b_1 x\right)}{b_2^{a_2} x^{a_2 - 1} \exp\left(-b_2 x\right)} \cdot \frac{\Gamma(a_2)}{\Gamma(a_1)}\right)$$

$$= (a_1 - a_2) \ln(x) + (-b_1 x + b_2 x) + \ln\left(\frac{\Gamma(a_2)}{\Gamma(a_1)}\right) + (a_1 \ln(b_1) - a_2 \ln(b_2)).$$

Thus, the integral becomes:

$$\mathrm{KL}(p \parallel q) = \int_0^\infty \frac{b_1^{a_1} x^{a_1 - 1} \exp\left(-b_1 x\right)}{\Gamma(a_1)} \left[(a_1 - a_2) \ln(x) + (b_2 - b_1)x + \ln\left(\frac{\Gamma(a_2)}{\Gamma(a_1)}\right) + (a_1 \ln(b_1) - a_2 \ln(b_2))\right] dx.$$

We now break this into four separate integrals.

$$I_1 = \int_0^\infty \frac{b_1^{a_1} x^{a_1 - 1} \exp\left(-b_1 x\right)}{\Gamma(a_1)}(a_1 - a_2) \ln(x) dx.$$

This integral can be solved using the properties of the Gamma distribution and the digamma function $\psi(a)$:

$$I_1 = (a_2 - a_1)\left(\ln(b_1) - \psi(a_1)\right),$$

where $\psi(a)$ is the digamma function, the derivative of the logarithm of the Gamma function.

$$I_2 = \int_0^\infty \frac{b_1^{a_1} x^{a_1} \exp\left(-b_1 x\right)}{\Gamma(a_1)}(b_2 - b_1) dx.$$

After performing the integration, we obtain:

$$I_2 = (b_2 - b_1)\frac{\Gamma(a_1 + 1)}{\Gamma(a_1)} \cdot \frac{1}{b_1} = (b_2 - b_1)a_1\frac{1}{b_1},$$

$$I_3 = \int_0^\infty \frac{b_1^{a_1} x^{a_1-1} \exp\left(-b_1 x\right)}{\Gamma(a_1)} \ln\left(\frac{\Gamma(a_2)}{\Gamma(a_1)}\right) dx.$$

Since this is a constant term, we can immediately evaluate it:

$$I_3 = \ln\left(\frac{\Gamma(a_2)}{\Gamma(a_1)}\right),$$

$$I_4 = \int_0^\infty \frac{b_1^{a_1} x^{a_1-1} \exp\left(-b_1 x\right)}{\Gamma(a_1)} (a_1 \ln(b_1) - a_2 \ln(b_2)) dx.$$

This integral simplifies to:

$$I_4 = a_1 \ln(b_1) - a_2 \ln(b_2).$$

Combining all the parts, the KL divergence between two Gamma distributions with rate parameters is:

$$\text{KL}(p \parallel q) = (a_2 - a_1)\left(\ln(b_1) - \psi(a_1)\right) + (b_2 - b_1)a_1 \frac{1}{b_1} + \ln\left(\frac{\Gamma(a_2)}{\Gamma(a_1)}\right) + a_1 \ln(b_1) - a_2 \ln(b_2).$$

## A.5 Predictive distribution

Through minimizing the negative log-marginal likelihood with observed training data, we can infer the distributions of the latent variables $q$, with which a predictive distribution can be derived. Given index set $\{i_1^p, \cdots, i_k^p, t_p\}$, we are to predict the corresponding value, we have:

$$p(y_p|\mathcal{D}) \simeq \int p(y_p|\{\mathbf{u}_{i_k^p,t_p}^k\}_{k=1}^K, \tau) q(\{\mathbf{u}_{i_k^p,t_p}^k\}_{k=1}^K) q(\tau) d(\{\mathbf{u}_{i_k^p,t_p}^k\}_{k=1}^K) d\tau$$

$$= \int\int \mathcal{N}(y_p|\mathbf{1}^\text{T}(\mathbf{u}_{i_1^p,t_p}^1 \circledast \cdots \circledast \mathbf{u}_{i_K^p,t_p}^k), \tau^{-1}) \prod_{k=1}^K q(\mathbf{u}_{i_k^p,t_p}^k) d(\mathbf{u}_{i_K^p,t_p}^K) q(\tau) d\tau$$

$$= \int\int \mathcal{N}(y_p|\mathbf{1}^\text{T}(\mathbf{u}_{i_1^p,t_p}^1 \circledast \cdots \circledast \mathbf{u}_{i_K^p,t_p}^K), \tau^{-1}) \prod_{k=1}^K \mathcal{N}(\mathbf{u}_{i_k^p,t_p}^k|\mathbf{g}^k(i_k^p, t_p), \sigma^2\mathbf{I}) d(\mathbf{u}_{i_k^p,t_p}^k) \text{Gamma}(\tau|\rho, \iota) d\tau$$

$$= \int\int \mathcal{N}(y_p|\mathbf{1}^\text{T}(\mathbf{u}_{i_1^p,t_p}^1 \circledast \cdots \circledast \mathbf{u}_{i_K^p,t_p}^K), \tau^{-1}) \mathcal{N}(\mathbf{u}_{i_1^p,t_p}^1|\mathbf{g}^1(i_1^p, t_p), \sigma^2\mathbf{I}) d(\mathbf{u}_{i_1^p,t_p}^1)$$
$$\prod_{k\neq 1} \mathcal{N}(\mathbf{u}_{i_k^p,t_p}^k|\mathbf{g}^k(i_k^p, t_p), \sigma^2\mathbf{I}) d(\mathbf{u}_{i_k^p,t_p}^k) \text{Gamma}(\tau|\iota, \iota) d\tau$$

$$= \int\int \mathcal{N}(y_p|(\underset{k\neq 1}{\circledast} \mathbf{u}_{i_k^p,t_p}^k)^\text{T} \mathbf{u}_{i_1^p,t_p}^1, \tau^{-1}) \mathcal{N}(\mathbf{u}_{i_1^p,t_p}^1|\mathbf{g}^1(i_1^p, t_p), \sigma^2\mathbf{I}) d(\mathbf{u}_{i_1^p,t_p}^1)$$
$$\prod_{k\neq 1} \mathcal{N}(\mathbf{u}_{i_k^p,t_p}^k|\mathbf{g}^k(i_k^p, t_p), \sigma^2\mathbf{I}) d(\mathbf{u}_{i_k^p,t_p}^k) \text{Gamma}(\tau|\rho, \iota) d\tau \qquad (28)$$

$$= \int\int \mathcal{N}(y_p|(\underset{k\neq 1}{\circledast} \mathbf{u}_{i_k^p,t_p}^k)^\text{T} \mathbf{g}^1(i_1^p, t_p), \tau^{-1} + \sigma^2 (\underset{k\neq 1}{\circledast} \mathbf{u}_{i_k^p,t_p}^k)^\text{T} (\underset{k\neq 1}{\circledast} \mathbf{u}_{i_k^p,t_p}^k))$$
$$\prod_{k\neq 1} \mathcal{N}(\mathbf{u}_{i_k^p,t_p}^k|\mathbf{g}^k(i_k^p, t_p), \sigma^2\mathbf{I}) d(\mathbf{u}_{i_k^p,t_p}^k) \text{Gamma}(\tau|\rho, \iota) d\tau$$

$$\vdots$$

$$= \int \mathcal{N}\big(y_p|\mathbf{1}^\text{T} \underset{k}{\circledast} \mathbf{g}^k(i_k^p, t_p), \tau^{-1} + \sigma^2 \sum_{j=1}^K (\underset{k\neq j}{\circledast} \mathbf{g}^k(i_k^p, t_p))^\text{T}(\underset{k\neq j}{\circledast} \mathbf{g}^k(i_k^p, t_p)) \text{Gamma}(\tau|\rho, \iota) d\tau$$

$$= \mathcal{T}\big(y_p|\mathbf{1}^\text{T} \underset{k}{\circledast} \mathbf{g}^k(i_k^p, t_p), \{\frac{\iota}{\rho} + \sigma^2 \sum_{j=1}^K (\underset{k\neq j}{\circledast} \mathbf{g}^k(i_k^p, t_p))^\text{T}(\underset{k\neq j}{\circledast} \mathbf{g}^k(i_k^p, t_p))\}^{-1}, 2\rho).$$

We found the prediction distribution follows the student's-t distribution.

## A.6 Closed Form of Predictive Distribution

After obtaining the variational posteriors of the latent variables, we can further derive the predictive distribution of the new data with arbitrary indexes. Given the index set $\{i_1^p, \cdots, i_k^p, t_p\}$ for prediction, we can obtain the variational predictive posterior distribution, which follows a Student's t-distribution (See Appendix A.5 for more details):

$$
\begin{aligned}
&p(y_p|\mathcal{D}) \sim \mathcal{T}(y_p|\mu_p, s_p, \nu_p), \\
&\mu_p = \mathbf{1}^\mathrm{T}[\underset{k}{\circledast}\mathbf{g}^k(i_k^p, t_p)], \quad \nu_p = 2\rho, \\
&s_p = \{\frac{\iota}{\rho} + \sigma^2 \sum_{j=1}^K [\underset{k \neq j}{\circledast}\mathbf{g}^k(i_k^p, t_p)]^\mathrm{T}[\underset{k \neq j}{\circledast}\mathbf{g}^k(i_k^p, t_p)]\}^{-1},
\end{aligned}
\tag{29}
$$

where $\mu_p$, $s_p$, $\mu_p$ is the mean, scale parameter and degree of freedom of the Student's t-distribution, respectively. The closed-form predictive distribution is a great advantage for the prediction process, as it allows us to do the probabilistic reconstruction and prediction with uncertainty quantification over the arbitrary continuous indexes.

# B  Additional experiment results

## B.1  Experiment settings: synthetic data

The CATTE was implemented with PyTorch [35] and `torchdiffeq` library (`https://github.com/rtqichen/torchdiffeq`). We employed a single hidden-layer neural network (NN) to parameterize the encoder. Additionally, we used two NNs, each with two hidden layers, for derivative learning and for parameterizing the decoder, respectively. Each layer in all networks contains 100 neurons. We set the dimension of Fourier feature $M = 32$, the ODE state $J = 5$ and the initial number of components of the latent factor trajectories $R = 5$. The CATTE was trained using Adam [36] optimizer with the learning rate set as $5e^{-3}$. The hyperparamters $\{a_r^0, b_r^0\}_{r=1}^R, c^0, d^0$ and initial values of learnable parameters $\{\alpha_r, \beta_r\}_{r=1}^R, \rho, \sigma^2, \iota$ are set to $1e^{-6}$ (so that all the initial posterior means of $\{\lambda_r\}_{r=1}^R$ equal 1). We ran 2000 epochs, which is sufficient for convergence.

## B.2  Experiment settings: real-world data

For THIS-ODE, we used a two-layer network with the layer width chosen from $\{50, 100\}$. For DEMOTE, we used two hidden layers for both the reaction process and entry value prediction, with the layer width chosen from $\{50, 100\}$. For LRTFR, we used two hidden layers with $100$ neurons to parameterize the latent function of each mode. We varied $R$ from $\{3, 5, 7\}$ for all baselines. For deep-learning based methods, we all used `tanh` activations. For FunBaT, we varied Matérn Kernel $\{1/2, 3/2\}$ along the kernel parameters for optimal performance for different datasets. We use ADAM optimizer with the learning rate tuned from $\{5e^{-4}, 1e^{-3}, 5e^{-3}, 1e^{-2}\}$.

## B.3  Rank learning curves of different datasets

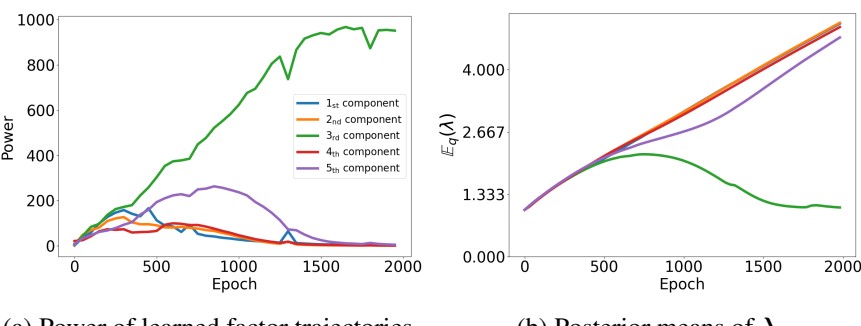

(a) Power of learned factor trajectories.      (b) Posterior means of $\boldsymbol{\lambda}$.

Figure 5: Rank learning curves of the synthetic data.

Fig. 5 plots the rank-learning curves during the gradient descent iterations of the synthetic data. That is, the evolutions of (a) the power of $R$ components of the estimated posterior mean of the factor trajectories [4] and (b) the values of estimated posterior mean of $\{\lambda_r\}_{r=1}^R$. Note that the power of $r$-th component of factor trajectories is conditioned by $\lambda_r$ (as shown (18) and *Remark 1*), and we plot the pair with the same color. One can see that as the epoch increases, the power of 4 components of the factor trajectories are forced to 0, which aligns with the increments of 4 corresponding $\lambda_r$s. And we can manually exclude the four components, which will not affect the final prediction results [5]. Only the third component ($r = 3$) is activated after convergence and $\lambda_3$ settles at a small value correspondingly. This indicates that our method successfully identifies the true underlying rank (i.e., 1) of the synthetic data while effectively pruning the other four components. Fig. (6) plots the rank-learning curves of the CA traffic, Server and SSF datasets respectively. In the same sense, we can infer that the revealed ranks of these three datasets are 5,7,7 respectively.

---

[4] We define the power of $r$-th component of factor trajectories as: $\sum_{k=1}^K \int_{i_k} \int_t (\mathbf{g}_r^k(i_k, t))^2 di_k dt$, which represents the contribution of the $r$-th component to the final output. The power can be approximated using $\sum_{n=1}^N \sum_{k=1}^K (\mathbf{g}_r^k(i_k^n, t_n))^2$.

[5] Our criterion for excluding the $r$-th rank is when $\mathbb{E}(\lambda_r)$ is large and power $\sum_{n=1}^N \sum_{k=1}^K (\mathbf{g}_r^k(i_k^n, t_n))^2$ is relatively small.

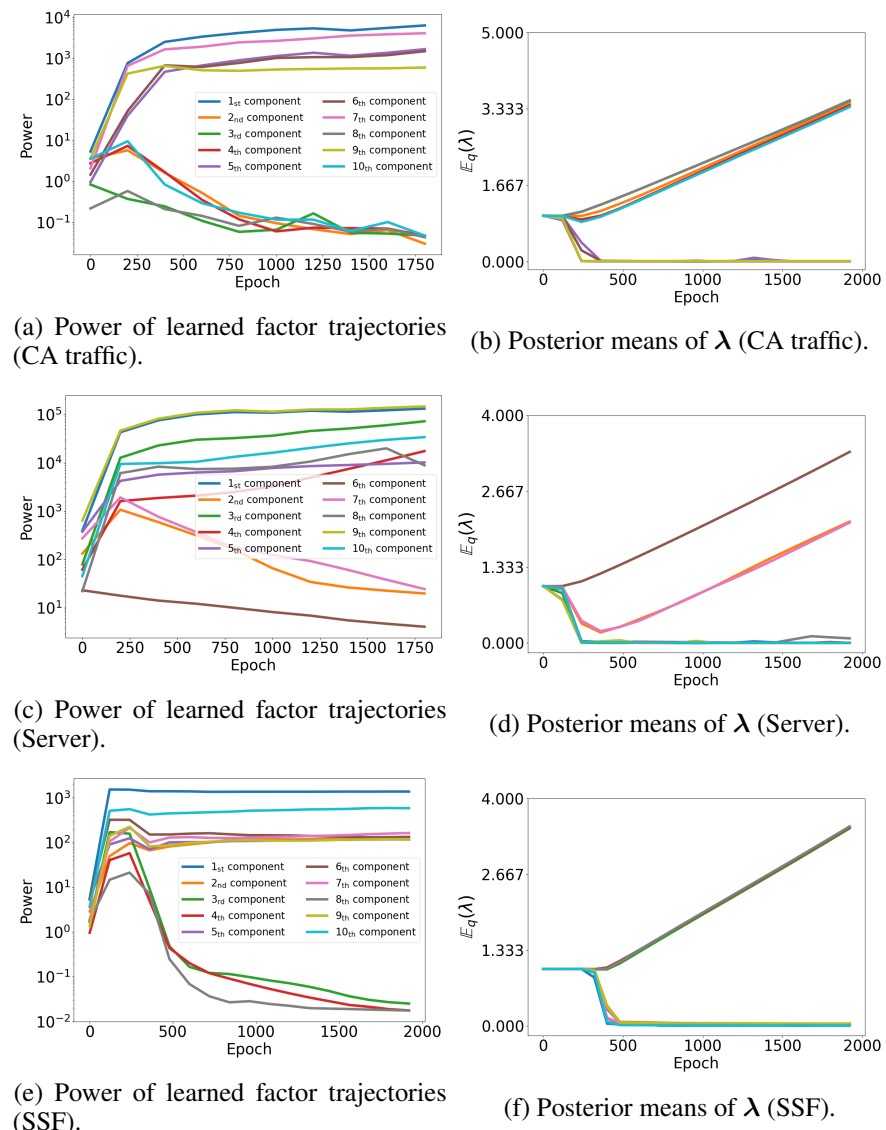

(a) Power of learned factor trajectories (CA traffic).

(b) Posterior means of $\boldsymbol{\lambda}$ (CA traffic).

(c) Power of learned factor trajectories (Server).

(d) Posterior means of $\boldsymbol{\lambda}$ (Server).

(e) Power of learned factor trajectories (SSF).

(f) Posterior means of $\boldsymbol{\lambda}$ (SSF).

Figure 6: Rank learning curves of three datasets.

## B.4 Noise robustness

Here, we show the robustness of our proposed FARD against the noise on the CA traffic dataset by comparing with the baselines with varying ranks on Tab. 4 and Tab. 5. Then, we show the robustness of CATTE against varing levels and types of noises on Tab. 6. The results shows that CATTE do not show significant performance degrade with varing levels of noises and even non-Gaussian noises, demonstating its robustness.

| | CATTE | CATTE w.o. FARD | LRTFR (R=5) | DEMOTE (R=5) |
|---|---|---|---|---|
| Noise Variance | **MAE** | | | |
| 0.5 | **0.134 ± 0.001** | 0.169 ± 0.022 | 0.213 ± 0.007 | 0.148 ± 0.003 |
| 1 | **0.182 ± 0.006** | 0.219 ± 0.006 | 0.305 ± 0.009 | 0.219 ± 0.034 |

Table 4: Extra experimental results on the robustness of functional automatic rank determination mechanism against the noise on the CA traffic dataset. The results were averaged over five runs.

|  | LRTFR (R=7) | DEMOTE (R=7) | LRTFR (R=10) | DEMOTE (R=10) | LRTFR (R=15) | DEMOTE (R=15) |
|---|---|---|---|---|---|---|
| Noise Variance | **RMSE** | | | | | |
| 0.5 | $0.484 \pm 0.023$ | $0,423 \pm 0.019$ | $0.469 \pm 0.053$ | $0.4375 \pm 0.005$ | $0.475 \pm 0.018$ | $0.455 \pm 0.005$ |
| 1 | $0.5475 \pm 0.011$ | $0.517 \pm 0.020$ | $0.621 \pm 0.038$ | $0.547 \pm 0.024$ | $0.708 \pm 0.013$ | $0.552 \pm 0.003$ |
| Noise Variance | **MAE** | | | | | |
| 0.5 | $0.224 \pm 0.019$ | $0.157 \pm 0.006$ | $0.232 \pm 0.012$ | $0.148 \pm 0.001$ | $0.245 \pm 0.011$ | $0.140 \pm 0.002$ |
| 1 | $03335 \pm 0.018$ | $0.209 \pm 0.020$ | $0.402 \pm 0.032$ | $0.312 \pm 0.013$ | $0.469 \pm 0.0135$ | $0.319 \pm 0.0025$ |

Table 5: Extra experimental results on the robustness of functional automatic rank determination mechanism against the noise on the CA traffic dataset. The results were averaged over five runs.

| Gaussion noise | **RMSE** | **MAE** |
|---|---|---|
| $\sigma^2$=0.05 | $0.056 \pm 0.004$ | $0.045 \pm 0.003$ |
| $\sigma^2$=0.10 | $0.090 \pm 0.005$ | $0.068 \pm 0.005$ |
| $\sigma^2$=0.15 | $0.112 \pm 0.006$ | $0.086 \pm 0.005$ |
| Laplacsian noise | **RMSE** | **MAE** |
| $\sigma^2$=0.05 | $0.065 \pm 0.006$ | $0.052 \pm 0.006$ |
| $\sigma^2$=0.10 | $0.094 \pm 0.008$ | $0.069 \pm 0.006$ |
| $\sigma^2$=0.15 | $0.117 \pm 0.07$ | $0.089 \pm 0.007$ |
| Poisson noise | **RMSE** | **MAE** |
| $\sigma^2$=0.05 | $0.064 \pm 0.004$ | $0.049 \pm 0.005$ |
| $\sigma^2$=0.10 | $0.096 \pm 0.010$ | $0.073 \pm 0.009$ |
| $\sigma^2$=0.15 | $0.115 \pm 0.012$ | $0.081 \pm 0.01$ |

Table 6: Extra experimental results on the robustness of functional automatic rank determination mechanism against various noise on the synthetic dataset. The results were averaged over five runs.

## B.5 Running time

Here we provide the comparisions of running time of different methods in Tab.7.

|  | CA traffic | Server Room | SSF |
|---|---|---|---|
| **THIS-ODE** | 283.9 | 144.8 | 158.9 |
| **NONFAT** | 0.331 | 0.81 | 0.67 |
| **DEMOTE** | 0.84 | 7.25 | 1.08 |
| **FunBaT-CP** | 0.059 | 0.027 | 0.075 |
| **FunBaT-Tucker** | 2.13 | 3.20 | 2.72 |
| **LRTFR** | 0.098 | 0.178 | 0.120 |
| **CATTE** | 0.227 | 0.83 | 0.247 |

Table 7: Per-epoch/iteration running time of different methods (in seconds).

## B.6 Hyperparameter Analysis

We first test the influence of the dimension of ODE latent state $J$ in Tab. 8. We observe that the final results are robust to the selection of $J$.

| $J$ | 6 | 8 | 10 | 16 |
|---|---|---|---|---|
| RMSE | **0.279** | 0.294 | 0.284 | 0.290 |
| MAE | 0.090 | 0.088 | **0.085** | 0.088 |

Table 8: Performance of CATTE under different $J$ on the CA traffic dataset. The results were averaged over five runs.

Then, we test influence of the hyper-parameters $a_r^0$ and $b_r^0$ in Tab. 8. We observe that the final results are robust to the selection of $a_r^0$ and $b_r^0$.

|  | $a_r^0 = b_r^0 = 1e^{-6}$ | $a_r^0 = b_r^0 = 1e^{-4}$ | $a_r^0 = 1e^{-6}, b_r^0 = 2e^{-4}$ |
|---|---|---|---|
| RMSE | $0.056 \pm 0.004$ | $0.056 \pm 0.003$ | $0.054 \pm 0.005$ |
| MAE | $0.045 \pm 0.003$ | $0.044 \pm 0.003$ | $0.044 \pm 0.004$ |

Table 9: Performance of CATTE under different $a_r^0$ and $b_r^0$ on the synthetic data experiment. The results were averaged over five runs.

## B.7 Additional visualization results with other baselines

Here, we show more visualization results of predictions on different methods and datasets on Fig. 7, Fig. 8 and Fig. 9. Our method yields qualitatively better visual results compared to the baselines.

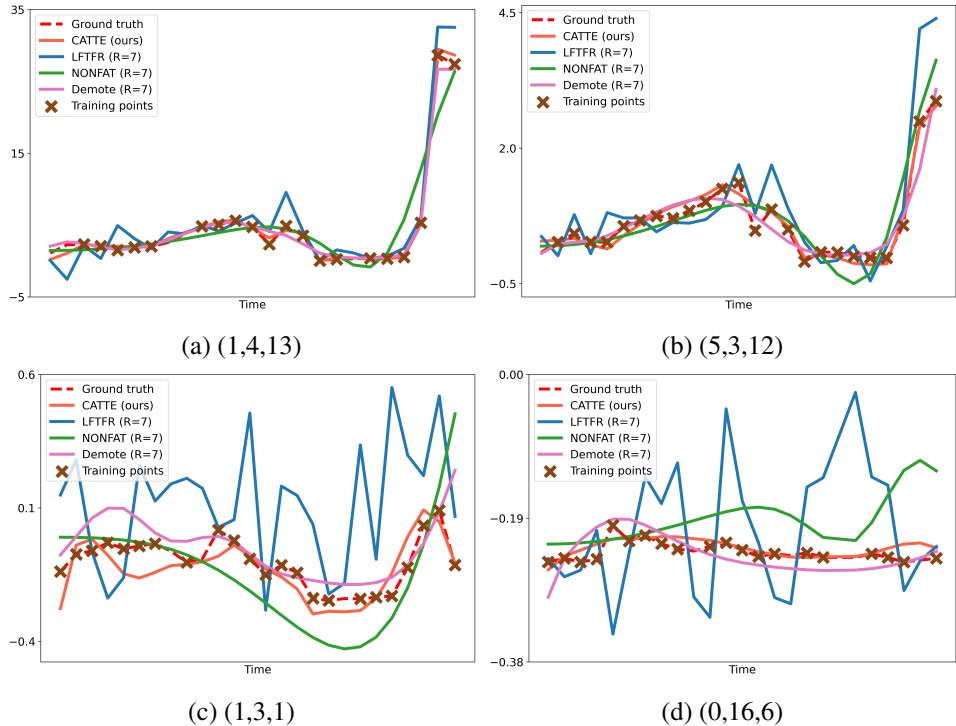

Figure 7: Additional visualization results on the *CA Traffic* dataset at different coordinates.

## B.8 Additional results on scalability

To further demonstrate the scalability of CATTE , we conducted synthetic experiments across varying configurations of timestamps×spatial points. All experiments employed the same network architecture, with $5\%$ of the total data points randomly selected for training. Therefore, the size of the training dataset ranges from $10^5$ to $10^7$. We hereby report the average training time (in seconds) per epoch in Table 10.

One can see that our method also scales well on the spatial mode since our method decouple each mode and convert the expanding spatial resulotion to the increase of the ODE states. For example, for the spatial resolution $50 \times 50$, we only need to solve $50 + 50 = 100$ ODE states. Likelywise, for the spatial resolution $200 \times 200$, we only need to solve $200 + 200 = 400$ ODE states. We will supplement the results in the latest version.

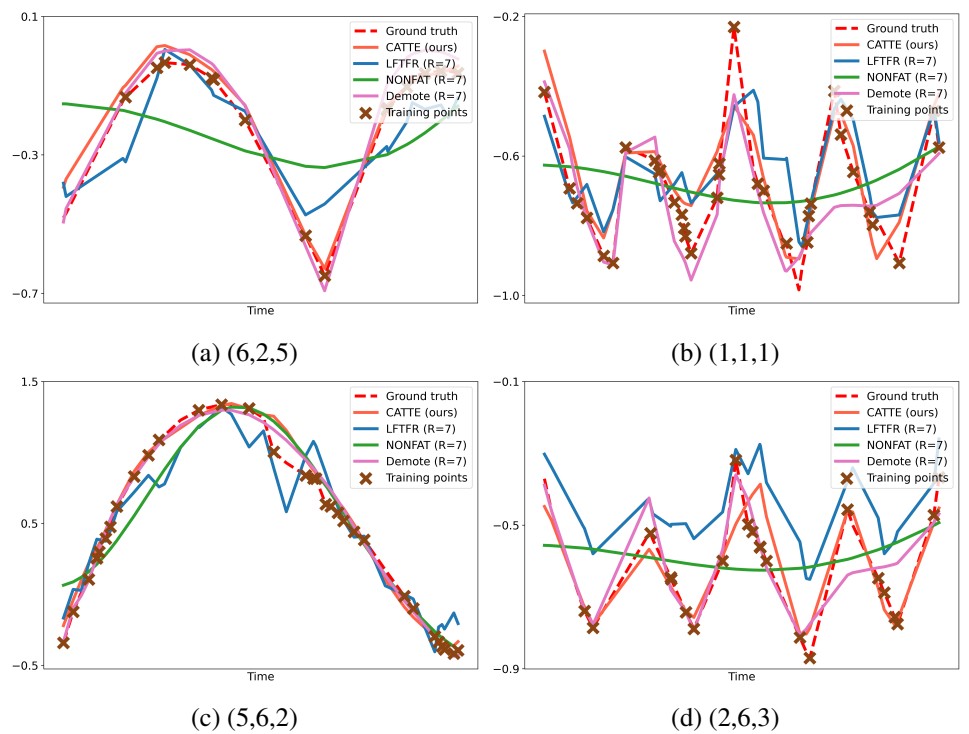

Figure 8: Additional visualization results on the *Server Room* dataset at different coordinates.

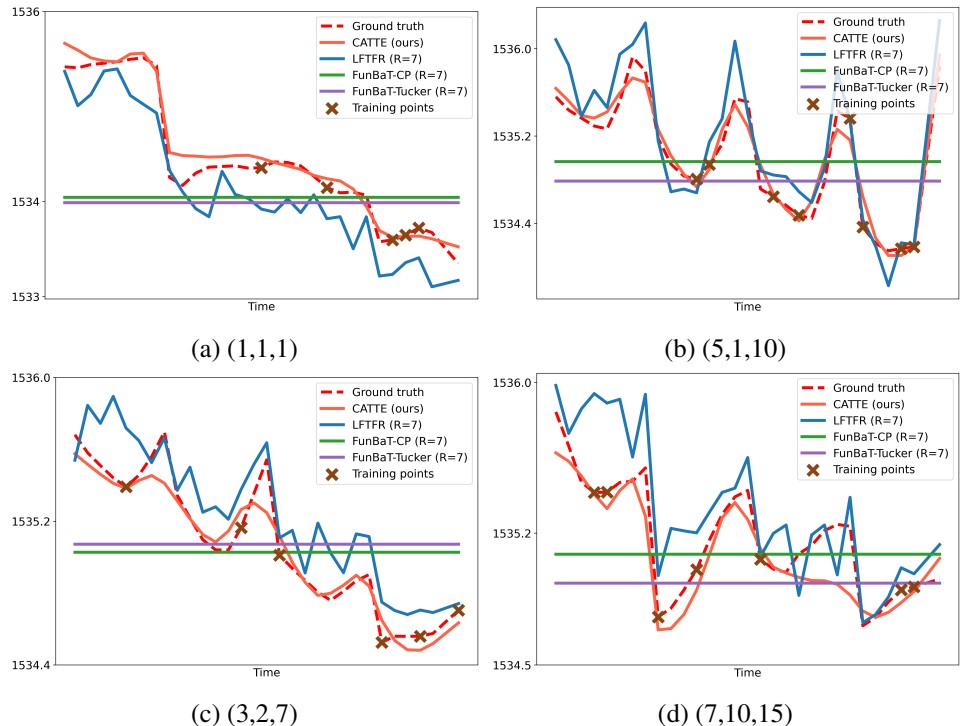

Figure 9: Additional visualization results on the *SSF* dataset at different coordinates.

## C   More Comments on the functional automatic rank determination mechanism

Our proposed functional automatic rank determination mechanism is different from previous method[14, 9, 8], which can not be directly applied for functional tensors. The differences are:

| timestamps  spatial resolution | $50 \times 50$ | $100 \times 50$ | $200 \times 50$ | $100 \times 100$ | $200 \times 200$ |
|:---:|:---:|:---:|:---:|:---:|:---:|
| $T = 100$ | 0.398 | 0.673 | 1.011 | 1.016 | 2.392 |
| $T = 200$ | 0.814 | 1.088 | 1.670 | 1.714 | 5.081 |
| $T = 500$ | 1.688 | 2.162 | 3.755 | 3.721 | 10.79 |

Table 10: Per-epoch/iteration running time on different size of tensors(in seconds).

**a) Derivation of ELBO:** Previous methods predefine separate latent factors and other latent variables (e.g., $\boldsymbol{\lambda}$) as variational parameters and then derive the ELBO. However, maintaining these parameters quickly consume memory as the dataset grows. Instead, we characterize the variational posterior mean of the latent factors using newly proposed continuous-indexed latent ODEs. This not only distinguishes our ELBO derivation from earlier methods but also overcomes scalability issues when handling large datasets.

**b) Optimization of ELBO:** In previous methods, seperate variational parameters are updated iterativelyadjusting one variable at a time while keeping the others fixed which limits the ability to leverage distributed computing resources. In contrast, we use functional parameterization of the posterior distribution and derive a closed-form ELBO that can be efficiently optimized using gradient descent. This allows all variational parameters to be updated simultaneously, making our method highly parallelizable and well-suited for GPUs.

Furthermore, previous methods struggle to infer out-of-bound data points while our model can handle them effectively.

Overall, CATTE integrates the strengths of deep learningscalability, powerful representations, and flexible architectureswith the benefits of Bayesian learning, including robustness to noise, uncertainty quantification, and adaptive model complexity, to offers an elegant solution for generalized temporal tensor decomposition.

# D   Impact Statement

This paper focuses on advancing temporal tensor decomposition techniques to push the boundaries of tensor decomposition. We are mindful of the broader ethical implications associated with technological progress in this field. Although immediate societal impacts may not be evident, we recognize the importance of maintaining ongoing vigilance regarding the ethical use of these advancements. It is crucial to continuously evaluate and address potential implications to ensure responsible development and application in diverse scenarios.

