# OpenReview forum: "Functional Complexity-adaptive Temporal Tensor Decomposition"
_NeurIPS.cc/2025/Conference — NeurIPS 2025 poster_

### Official Review · Reviewer_q4xE · 2025-06-28

**Clarity:** 3
**Significance:** 3
**Originality:** 2
**Rating:** 5
**Confidence:** 4

**Summary:**

The paper proposes CATTE, a Bayesian CP-style model for generalized temporal tensors whose modes may all be continuous-indexed. CATTE (i) represents spatial coordinates through learnable Fourier features and evolves factors with a latent Neural ODE, (ii) places a dimension-wise Gaussian-Gamma prior over the entire family of factor trajectories, and (iii) derives a fully closed-form evidence-lower-bound that admits sampling-free variational inference. Experiments on one synthetic and three real-world data sets (CA-Traffic, Server-Room and Pacific Sound-Speed Field) show that CATTE outperforms five recent baselines in RMSE/MAE while automatically discovering the ground-truth rank, and exhibiting robustness to several noise types.

**Questions:**

Could you articulate a specific theoretical insight or property that cannot be achieved by any pair-wise combination of Fourier features, neural-ODE dynamics, and ARD sparsity alone?
In other words, what new capability or guarantee emerges uniquely from the full CATTE synthesis?

For each dataset, what is the total number of trainable parameters in CATTE versus the baselines?
If you downscale CATTE so that its parameter budget matches that baseline, how do RMSE/MAE and wall-clock convergence change?

Beyond time horizon, how does CATTE's runtime scale with the number of spatial points?
A synthetic experiment sweeping both dimensions (timestamps × spatial points) would help practitioners judge feasibility.

**Ethical Concerns:**

["NO or VERY MINOR ethics concerns only"]

**Final Justification:**

The reviewers addressed my concerns, and I raised my score to accept.

**Limitations:**

Yes

**Quality:**

3

**Strengths And Weaknesses:**

Strengths

S1. The paper is technically solid: it frames the decomposition in a fully Bayesian manner and derives an evidence-lower-bound whose every term is analytic.
S2. The motivation and related-work sections clearly position the contribution within the temporal-tensor literature.
S3. Extending automatic-rank ideas to functional factor trajectories and producing a closed-form ELBO for this setting is, to the best of my knowledge, novel and likely to stimulate further research in probabilistic tensor analysis.

Weaknesses
W1. (Limited methodological novelty)
Although the paper delivers a well-engineered system, each of its core building blocks (Fourier positional encodings, neural-ODE dynamics, and an automatic-rank-determination sparsity prior) are already well established in the literature. The contribution therefore lies chiefly in how these standard components are combined, rather than in a fundamentally new modelling principle or theoretical insight.

W2. (Parameter efficiency and fairness of comparison)
Equation (5) defines every latent factor as a trajectory u^k(i_k, t) that is jointly conditioned on the continuous spatial index i_k and on the timestamp t. This modelling choice implies that CATTE's learnable parameter set grows proportionally with #spatial points x #time-stamps x R. The paper does not report the resulting parameter counts, whereas the baselines rely on time-invariant factors or low-order spline/GP coefficients. Only per-iteration runtimes are given (Table 6), so the accuracy gains in Table 1 may partly reflect CATTE's much larger capacity rather than genuine efficiency. A like-for-like parameter budget (or test-error curves versus model size) would be necessary for a fair comparison.

W3. (Limited empirical scale)
Evaluation is limited to three modest-scale datasets. The benchmarks contain roughly 10,000 observed entries and at most 34 timestamps. No experiment tests very large (>10^6) data, so the linear-time claim remains speculative for industrial-scale workloads.

W4. (Presentation issues)
Several typographical errors (e.g., “Exisiting” -> “Existing”: line 48, “Give” -> “Given”: line 148, “contains” -> “containing”: line 281, “demonstating” -> “demonstrating”: line 559, etc.) distract from the narrative.

---

> ### Author Rebuttal · Authors · 2025-07-29
>
> We sincerely appreciate your thoughtful feedback and  recognition of our work. Below, we address the comments (C: comment, R: response):
>
>
> >C1: Concerns about the methodological novelty.
>
>
> R1:  We list the comparison of CATTE and other main  methods in the following table:
> |   Property / Methods   |  **CATTE** | ThisODE|Functional tensor methods | Bayesian neural ODEs | Factor-evolving tensor methods |
> |:-----:|:-----:|:-----:|:-----:|:-----:|:-----:|
> |Continuous  dynamics modeling | **Yes** | Yes | Partial (tend to underfit the complex dynamics) | Yes|Yes|
> |Continuous-indexed  modes modeling | **Yes** | No |Yes| No|No|
> |Scability | **High** | Low |High| Low|Low|
> |Complexity self-adpation| **Yes** | No |No| No|No|
> |Uncertainty-aware| **Yes** | No |Partial| Partial|No|
>
> For  more details about our contributions, pls refer to   R1 of Reviewer MW7k.
>
>
> >C2: This modelling choice implies that CATTE's learnable parameter set grows proportionally with #spatial points x #time-stamps x R.
>
> R2: We would like to clarify that the number of learnable parameters in CATTE **does not** grow proportionally with the number of spatial points $\times$ time-stamps. For each non-temporal mode, we employ a fixed network for continuous index representation learning. Similarly, for the temporal mode, we use a fixed network to learn the derivatives. Our model only grows  with $R$. As a result, our model can handle data with arbitrarily  resolution while maintaining parameter efficiency.
>
> >C3:The paper does not report the resulting parameter counts, whereas the baselines rely on time-invariant factors or low-order spline/GP coefficients.
>
> R3: Thanks for the comment, we report the parameters in R7.
>
>
> >C4:Limited empirical scale. Evaluation is limited to three modest-scale datasets.
>
> R4: Thank you for the comment. The three real-world datasets we use are widely adopted in this literature [6,15,28], ensuring fair comparison with prior work.
>
> To address concerns about scalability, we additionally conducted experiments on a much larger real-world dataset, SSF-large, which spans four dimensions: 38 latitudes × 76 longitudes × 50 depths × 100 timestamps — yielding over $10^7$ data points in total. We randomly selected $10^6$ coordinate–value pairs for training and the rest for testing. This experiment is approximately **100× larger** in scale compared to those in the main section. So the amount training and test data exceed $10^6$. To handle the large data, we increase the size of all model and make their parameters comparable.
>
> We report the RMSE/MAE, model parameters and training time per epoch in the following:
> |     | CATTE （R=10） | ThisODE (R=10)| DEMOTE (R=10)| LRTFR (R=10)|FunBaT-CP (R=10)|FunBaT-Tucker (R=10)|NONFAT (R=10)|
> |:-----:|:-----:|:-----:|:-----:|:-----:|:-----:|:-----:|:-----:|
> | RMSE | **0.406 $\pm$ 0.003**| N/A |N/A | 0.713 $\pm$ 0.0781|0.819 $\pm$ 0.134|N/A|9.812±0.001|
> | MAE | **0.319 $\pm$ 0.004** | N/A |N/A |0.547 $\pm$ 0.0435|0.622 $\pm$ 0.100|N/A| 8.761±0.001|
> | Parameters | 420K | 428K |650K|462K|N/A|N/A|N/A|
> | Time per Epoch  | 9.38s | >30min |>30min |9.96s|3.70s|>30min|67.2s|
>
> Our method is superior to other baselines in accuracy.
> We find that among baselines only LRTFR and FunBaT-CP  scales well to the large dataset. NONFAT does not produce meaningful trajectories. For those methods whose per‑epoch runtime is prohibitively long, we do not report the final results.
>
> To further demonstrate the scalability of our method, we conducted large-scale experiments on synthetic data, as detailed in R8.
>
>
>
>
> We will include this additional large-scale experiment and its results in the latest version
>
>
>
>
>
>
>
> >C5:Presentation issues
>
> R5: Thank you for the corrections. We will thoroughly review the manuscript and address all typographical errors we found in the revised version.
>
>
> >C6: Could you articulate a specific theoretical insight or property that cannot be achieved by any pair-wise combination of Fourier features, neural-ODE dynamics, and ARD sparsity alone? In other words, what new capability or guarantee emerges uniquely from the full CATTE synthesis?
>
> R6: Thank you for your insightful question. We provide the
>
>
> | Combination Type           | Captures                                  | Missing Capability                                                    |
> |----------------------------|-------------------------------------------|------------------------------------------------------------------------|
> | **Fourier + Neural-ODE**   | Continuous-indexed temporal  dynamics modeling  | Lacks adaptive sparsity; may overfit noisy or irrelevant components; Lack uncertainty quantification.    |
> | **Fourier + FARD**   | Continuous-indexed tensor modeling with adaptive complexity control | No temporal modeling; fails to capture complex dynamic dependencies           |
> | **Neural-ODE + FARD**| Sparse latent temporal dynamics            | Cannot effectively represent  continuous-indexed temporal signals or learn spatial correlations; Scalibility issue for large dataset may arise. |
>
>
> In contrast, **CATTE** synergistically combines:
> - The expressivity of **Fourier** features for positional representation,
> - The continuity and strong representitive power of **Neural-ODE** latent dynamics,
> - The parsimony induced by **FARD** sparsity,
>
> resulting an efficient and scalable model for reconstructing temporal tensors from sparse, noisy, and continuously indexed observations.
>
> >C7:For each dataset, what is the total number of trainable parameters in CATTE versus the baselines? If you downscale CATTE so that its parameter budget matches that baseline, how do RMSE/MAE and wall-clock convergence change?
>
>
> R7: The total number of trainable parameters in CATTE  versus the baselines on  CA Traffic, ServerRoom and  SSF datasets are:
> |   Parameters   | CATTE （R=10） | ThisODE (R=10)| DEMOTE (R=10)| LRTFR (R=10)|
> |:-----:|:-----:|:-----:|:-----:|:-----:|
> | - | 90K | 89K |105K|91K|
>
> They are comparable since the number of layers (2) and  layer width (100) are set to same as stated in Appendix B.2.  FunBaT and NONFAT are non-parametric GP-based model and do not have trainable parameters.
>
> We also varying the rank of the baslines, and when $R=3$, the deep-learning based baselines are:
>
> |   Parameters    | ThisODE (R=3)| DEMOTE (R=3)| LRTFR (R=3)|
> |:-----:|:-----:|:-----:|:-----:|
> | -  | 64K |83K|67K|
>
>
> We downscale CATTE by setting $R=3$, the number of parameters is 69K. We reconduct the experiment, the results are:
> |    CATTE(R=3)    | CA traffic | Server Room | SSF|
> |:-----:|:-----:|:-----:|:-----:|
> | RMSE | 0.308 $\pm$ 0.024 | 0.107 $\pm$ 0.009| 0.435 $\pm$ 0.020 |
> |  MAE   | 0.113 $\pm$ 0.011 | 0.084 $\pm$ 0.005 | 0.033 $\pm$ 0.017 |
>
> We can observe  the increase in RMSE/MAE, which is primarily due to underfitting of the signal. Nevertheless, CATTE still outperforms the baseline methods. The convergence is slightly faster than the non-downscaled version of CATTE.
>
> In practice, CATTE can be manually configured with sufficient model parameters, and it will adaptively adjust its parameters to fit the data effectively. We will clarify these in the latest version
>
>
>
>
> >C8: Beyond time horizon, how does CATTE's runtime scale with the number of spatial points? A synthetic experiment sweeping both dimensions (timestamps × spatial points) would help practitioners judge feasibility.
>
> R8: Thank you for your suggestion. To demonstrate the scalability of our method, we conducted synthetic experiments using the same synthetic function as in Eq. (20), across varying configurations of timestamps $\times$ spatial points. All experiments employed the same network architecture, with $5\%$ of the total data points randomly selected for training. As a result, the size of the training dataset ranges from $10^5$ to $10^7$. Consistent with Fig. 4, we report the average training time (in seconds) per epoch.
>
>
>
> |   timestamps \ spatial resolution   | $50 \times 50$  | $100 \times 50$ | $200 \times 50$| $100 \times 100$|$200 \times 200$|
> |:-----:|:-----:|:-----:|:-----:|:-----:|:-----:|
> | T=100 | 0.398 | 0.673 |1.011|1.016|2.392|
> | T=200  | 0.814| 1.088|1.670|1.714|5.081|
> | T=500 | 1.688 | 2.162 |3.755|3.721|10.79|
>
>
>
> One can see that our method also scales well on the spatial mode since our method decouple each mode and convert the expanding spatial resulotion to the increase of the ODE states. For example, for the spatial resolution $50\times 50$, we only need to solve $50+50=100$ ODE states. Likelywise, for the spatial resolution $200\times 200$, we only need to solve $200+200=400$ ODE states. We will supplement the results in the latest version.

---

> > ### Comment · Area_Chair_f3fV · 2025-08-06
> > **Can you please respond to the authors?**
> >
> > Dear Reviewer,
> >
> > Can you please respond to the authors?
> >
> > Best,
> > The AC

---

### Official Review · Reviewer_q3bj · 2025-06-30

**Clarity:** 2
**Significance:** 2
**Originality:** 2
**Rating:** 4
**Confidence:** 3

**Summary:**

This paper proposes CATTE, a method for temporal tensor decomposition that handles tensors with continuous indices across all modes. The key innovation is combining neural ODEs with automatic rank determination for functional temporal tensors.

**Questions:**

1. Please discuss and clarify the difference and improvement over existing methods like DEMOTE and FunBaT
2. It seems to the ELBO derivation problems (Eq. 17) has dimensional inconsistencies. It needs clarification - what does vec() of a matrix plus identity $\mathbf{I}$ mean here? Similar issues raise in other places (e.g., Eq. 24)

**Ethical Concerns:**

["NO or VERY MINOR ethics concerns only"]

**Final Justification:**

Authors addressed some concerns of mine. However the kitchen sink approach (in my view) still affects my evaluation. I will maintain the score

**Limitations:**

Yes. In supplementary

**Quality:**

3

**Strengths And Weaknesses:**

**Strengthens**

1. Comprehensive baselines: Compares against 6 state-of-the-art methods spanning temporal (NONFAT, DEMOTE), functional (FunBaT, LRTFR), and other approaches
2. Consistent improvements: Shows great RMSE reduction across datasets, with particularly strong performance on spatialtemporal ones
3. Computational efficiency: Linear scaling with time series length
4. Existing methods either handle rank determination or continuous indices, but not both simultaneously. The solution is mathematically sound and showed improvement.

**Weaknesses**
1. Individual components (Neural ODEs, Bayesian rank selection, functional tensors) all exist. Thus, contribution appears largely incremental - combining existing techniques
2. Fourier feature dependency: Choice may not be optimal for all data type; What if sparse data. In practice, tensor data is very sparse. Sparse basis functions might be helpful.
3. Hyperparameter analysis is not enough. For example, having three sets of $a$ and $b$ for Gamma on CA traffic can't conclude the method auto rank selection is robust to initialization.
4. Missing analysis of related work in Bayesian Neural ODEs literature

---

> ### Author Rebuttal · Authors · 2025-07-29
>
> We thank the reviewer for the careful review! We address the comments below:( C: comment; R: response)
>
>
>
> >C1: Individual components (Neural ODEs, Bayesian rank selection, functional tensors) all exist. Thus, contribution appears largely incremental - combining existing techniques
>
> R1: We appreciate the reviewer’s concern. However, many works, including the many references in Related Work, also use existent techniques, via their novel application and combination, to address the problem of interest. we believe such works are still valuable contributions to the relevant fields.
>
> Our problem setting, which reconstructing temporal tensors from sparse, noisy, and continuously indexed observations is new and is practically important in environmental and scientific data, but remains largely unaddressed in prior literature.
> To tackle this challenge, we combine three componants and also propose  **methodological innovations** to ensure the compatibility and effectiveness  of the combinations:
> (1) A  continuous-indexed neural ODEs-based tensor decomposition framework  for handling continuously indexed temporal tensor data with efficiency-driven  approach that scales our model to large datasets. (2) A deep characterization of the variational posterior and new derivation of close-form ELBO to ensure both  effectiveness and scalibility.  **To the best of our knowledge, we are the first to  achieve continuous-indexed temporal tensor modeling,  scale ODE-based tensor decomposition to large datasets, and  enable automatic rank determination in functional tensors.** We list the comparison of CATTE
> and other main  methods in the following table:
> |   Property / Methods   |  **CATTE** | ThisODE|Functional tensor methods | Bayesian neural ODEs | Factor-evolving tensor methods |
> |:-----:|:-----:|:-----:|:-----:|:-----:|:-----:|
> |Continuous  dynamics modeling | **Yes** | Yes | Partial (tend to underfit the complex dynamics) | Yes|Yes|
> |Continuous-indexed  modes modeling | **Yes** | No |Yes| No|No|
> |Scability | **High** | Low |High| Low|Low|
> |Complexity self-adpation| **Yes** | No |No| No|No|
> |Uncertainty-aware| **Yes** | No |Partial| Partial|No|
>
> For much more details, pls refer to   R1 of Reviewer MW7k.
> We have validated our method across a wide range of  datasets, demonstrating both strong empirical performance and broad applicability.
>
>
>
> >C2:Fourier feature dependency: Choice may not be optimal for all data type; What if sparse data. In practice, tensor data is very sparse. Sparse basis functions might be helpful.
>
> R2: Thank you for your comment. We would like to clarify that the Fourier features in our work are used  for positional embedding, not for representing the data itself. Besides, our model is specifically designed to handle sparse tensor data by effectively capturing spatial and temporal continuity.  For example, the observation rate is 7.8% for the CA Traffic dataset, 2.6% for ServerRoom, and 11% for SSF.
>
>
> >C3: Robustness of auto rank selection to initialization.
>
> R3: In the objective function (16), $a_{r}^0$ and $b_{r}^0$ only affect the KL term, KL$(q(\lambda) \| p(\lambda))$. Following common practice in previous Bayesian CP works, we set them to small values to define a non-informative prior. This choice has minimal impact on our proposed method.
>
>
> >C4:Missing analysis of related work in Bayesian Neural ODEs literature
>
> R4: Thank you for the suggestion! We will include the following review of the literature on Bayesian Neural ODEs and clarify how our work differs from existing approaches.
>
> "[B1] combines a variational autoencoder with neural ODE dynamics, replacing the encoder with an ODE-RNN to handle irregularly-sampled time series. [B2] combines a continuous-time GRU with a Bayesian update mechanism to model evolving latent dynamics in  irregularly‑sampled time series. However, these Bayesian Neural ODEs are primarily designed for multivatiate time series, which is not applicable in our setting."
>
> [B1] Rubanova, Yulia, Ricky TQ Chen, and David K. Duvenaud. "Latent ordinary differential equations for irregularly-sampled time series." Advances in neural information processing systems 32 (2019).
>
> [B2] De Brouwer, Edward, et al. "GRU-ODE-Bayes: Continuous modeling of sporadically-observed time series." Advances in neural information processing systems 32 (2019).
>
>
>
>
>
> >C5: the difference and improvement over existing methods like DEMOTE and FunBaT
>
> R5: Thank you for your comments. The differences are:
> | | FunBaT | DEMOTE |CATTE|
> |:------:|:------:|:------:|:------:|
> | Formulation  |fully factorized and decoupled: $\boldsymbol{1}^{\text{T}}[\underset{k}{\circledast}\mathbf{u}^k(i_k) \circledast \mathbf{u}^{\text{Temporal}}(t) ]$|temporally coupled:$g([\mathbf{u}^k_{i_1}(t), \cdots, \mathbf{u}^k_{i_K}(t)])$, where $g$ is MLP. |Functional factor traj:$\boldsymbol{1}^{\text{T}}[\underset{k}{\circledast}\mathbf{u}^k(i_k,t) ]$.|
> | Model & Inference | Gaussian process(GPs) + Bayesian message passing|    Multi-Partite Graphs with diffusion process and  reaction process + Gradient descent on maximum a posteriori (MAP)|Continuous-indexed latent neural ODEs + Gradient descent on analytical ELBO|
> | Hyperparas. | Requires specifying tensor rank and kernel parameters.	|Requires specifying tensor rank.  |Automatic rank tuning.|
>
> The improvements are:
>
>
>
> * The fully factorized form of FunBaT can underfit cases where time mode is complex and naturally coupled with others. DEMOTE requires the discrete factors to be pre-specified and does not support continuous-indexed modeling. Moreover, the evolution of its factors depends on aggregating information from all modes, causing the number of parameters to grow **exponentially** with the number of spatial points. This significantly limits its scalability. **In contrast, CATTE effectively captures time-varying patterns by modeling factor trajectories jointly, and demonstrates excellent scalability, as shown in R4&R8 for Reviewer q4xE.**
>
> * FunBaT reduces computational cost by converting GPs into state-space priors via SDEs, but this limits kernel choices to stationary ones (e.g., Matérn), reducing expressiveness. Its Bayesian message-passing inference is length. DEMOTE uses MAP estimation, which lacks uncertainty quantification and is more sensitive to noise. **In contrast, CATTE  offers better architectural flexibility, enabling richer representations than FunBaT and adopts a Bayesian framework with a closed-form ELBO, enabling efficient gradient-based optimization, uncertainty quantification, and improved robustness to noise.**
>
>
> * FunBaT and DEMOTE requires to pre-set more hyperparameters (tensor rank, kernel paras) than CATTE, making fine-tuning more challenging.
>
> n summary, CATTE integrates the strengths of both FunBaT and DEMOTE while overcoming their respective limitations. Moreover, unlike these two methods, CATTE supports complexity adaptation. We will clarify this in the latest version
>
>
> >C6:It seems to the ELBO derivation problems (Eq. 17) has dimensional inconsistencies. It needs clarification - what does vec() of a matrix plus identity $\mathbf{I}$ mean here? Similar issues raise in other places (e.g., Eq. 24)
>
>
> R6: Thanks for the question. We would like to clarify that $\text{vec}\big( \mathbf{g} \mathbf{g}^{T} + \sigma^2 \mathbf{I} \big)$ of  Eq. 17 first performs the addition of two matrices of the same shape, and then vectorizes the result. We will improve the clarity in the latest version.

---

> > ### Comment · Reviewer_q3bj · 2025-08-06
> >
> > Thank authors for the detailed rebuttal and effort. I appreciate your clarifications. However, I respectfully maintain my original score. In particular, the concern regarding the complexity of the study design have not been sufficiently addressed. This aspect remains a significant factor affecting my evaluation.

---

> > > ### Author Response · Authors · 2025-08-06
> > >
> > > Thanks for your feedback!

---

### Official Review · Reviewer_MW7k · 2025-07-02

**Clarity:** 4
**Significance:** 3
**Originality:** 3
**Rating:** 4
**Confidence:** 3

**Summary:**

In this work, the authors try to solve the problem of temporal tensor decomposition with continuous indices in all modes. The CATTE algorithm is developed by employing the encoder-decoder structure and neural ODE to model the factor trajectories. An automatic tensor rank determination is utilized. A sampling-free variational inference algorithm with a closed-form ELBO is proposed. Experimental results on both synthetic and real-world datasets demonstrate the desired performance of the proposed method.

**Questions:**

1. The main concern of this paper is the limited novelty of the proposed method. As the encoder-decoder structure and neural ODE are common methods for modeling dynamical systems, the current algorithm seems to be an application to the temporal tensor decomposition domain (Specifically, the ODE has already been applied to temporal tensor decomposition [6][28]). Also, the automatic tensor rank determination method has already been developed. The author should give a clear statement on the contribution of the proposed method, such as a more comprehensive related work on encoder-decoder structure and neural ODE, and a detailed list of contributions for the proposed method.

2. How does CATTE scale with the size or dimension of tensors (e.g., 5-order or higher-order tensors and $100\times100\times500$ tensors)? Are there computational bottlenecks, especially with the ODE solvers or variational inference?

3. As the author shows the robustness to different noise distributions, it is interesting how about the robustness when the data contains missing entries.

4. Are there any smoothness (stationary) assumptions on the temporal dynamics of the tensor? How is the performance of the CATTE if there is a sudden change?

5. It seems the ref. [28] and ref. [34] are the same. Please check if there are some mistakes.

**Ethical Concerns:**

["NO or VERY MINOR ethics concerns only"]

**Final Justification:**

Based on the authors‘ responses, I would keep my score as borderline accept.
The main concerns, questions 1 and 2, are properly addressed. The author gives an additional table to highlight the advantages of the proposed method compared with existing methods. Further, the experiments on large tensors are carried out, demonstrating the scalability of the proposed method.

**Limitations:**

Yes

**Quality:**

3

**Strengths And Weaknesses:**

Strength:
1. A new method that generalizes the modeling the temporal tensor data with continuous indexes in all modes.
2. Experiments show good performance and robustness across diverse datasets.

Weakness:
1. The novelty is limited as the encoder-decoder structure, the neural ODE, and the automatic tensor rank determination method have already been used in modeling dynamic systems or tensor decomposition.
2. The scales and dimensions of the tensor data in experiments are relatively small.

---

> ### Author Rebuttal · Authors · 2025-07-29
>
> Thanks for the careful review! We address the comments below (C: comment, R: response)
>
>
> >C1:About the novelty and contribution
>
> R1: We appreciate the reviewer’s concern. **Our main contribution is to develop an efficient and scalable solution for reconstructing temporal tensors from sparse, noisy, and continuously indexed observations**. This setting  is both *practically significant and technically underexplored*.
>
> We select three complementary components, i.e., Fourier features, Neural-ODEs, and ARD sparsity, as they are each well-suited to different aspects of this challenging task. However, we emphasize that simply combining these components  does not staightforwardly lead to a functioning model.
>
> To address this, we propose several methodological innovations to ensure their **compatibility, integration, and effectiveness** within a unified Bayesian  framework. These design choices are key to enabling scale and robust performance under sparse, irregular and noisy data conditions.
>
>
>
> We are to highlight our novelty and contribution:
>
> **a) Continuous-indexed latent ODE:** We encodes continuous spatial indexes as learnable Fourier features and employs neural ODEs in latent space to learn the temporal trajectories of factors, which enables a new continuous-indexed dynamics modeling.   We also propose an efficiency-driven approach that concatenates the states of multiple indices across different modes and updates them simultaneously using a single ODE solver, thereby accelerating the learning of latent ODE states.
>
> Our framework fundamentally differs from related ODE-based methods [A1, 6, 8], all of which lack support for continuous-indexed temporal tensor modeling. Specifically, [A1] combines a variational autoencoder with an ODE model to handle multivariate time series.  [6] uses time-invariant latent factors as conditional inputs to an ODE. [28] employs a multipartite graph to model cross-mode interactions. Although both target temporal tensors, they also face scalability limitations.
>
>
>
>
> **b) Functional tensor rank automatic determination:** To the best of our knowledge, CATTE is the first work to achieve automatic rank determination in functional tensors, while eliminating the scalibitly and representation issues of previous methods [8,9]. The differences are:
> * Derivation of ELBO: Previous  methods [8,9] predefine separate latent factors and other latent variables (i.e., $\boldsymbol{\lambda},\tau$) as variational parameters and then derive the ELBO. However, maintaining these parameters quickly consume memory as the size of  tensor grows.
> In CATTE, we replace separate latent factors with latent factor functions and characterize their variational posterior mean using the newly proposed continuous-indexed latent ODEs. This effectively resolves the scalability issues encountered when handling large datasets.  Consequently, we contribute to derivate a new closed-form ELBO.
>
> * Optimization of ELBO: In previous methods [8,9], seperate variational parameters are updated iteratively—adjusting one variable at a time while keeping the others fixed— which limits the ability to leverage distributed computing resources.
> In contrast, CATTE use functional parameterization of the posterior distribution. It improves the representive power of posterior estimation. Also, through it, a closed-form ELBO can be derived and can be efficiently optimized using gradient descent. This allows all variational parameters to be updated simultaneously, making our method highly parallelizable and well-suited for GPUs.
>
>
> Overall, CATTE  integrates deep learning’s expressive power and architectural flexibility with Bayesian learning’s robustness, uncertainty quantification, and adaptive complexity, providing an effective and principled solution for the target task. **We list the comparison of CATTE
> and other main  methods in the following table:**
> |   Property / Methods   |  **CATTE** | ThisODE|Functional tensor methods | Bayesian neural ODEs | Factor-evolving tensor methods |
> |:-----:|:-----:|:-----:|:-----:|:-----:|:-----:|
> |Continuous  dynamics modeling | **Yes** | Yes | Partial (tend to underfit the complex dynamics) | Yes|Yes|
> |Continuous-indexed  modes modeling | **Yes** | No |Yes| No|No|
> |Scability | **High** | Low |High| Low|Low|
> |Complexity self-adpation| **Yes** | No |No| No|No|
> |Uncertainty-aware| **Yes** | No |Partial| Partial|No|
>
>  We will supplement the above dusscussions  in our paper.
>
> [A1] Rubanova, Yulia, Ricky TQ Chen, and David K. Duvenaud. "Latent ordinary differential equations for irregularly-sampled time series." Advances in neural information processing systems 32 (2019).
>
>
>
> >C2:How does CATTE scale with the size or dimension of tensors (e.g., 5-order or higher-order tensors and  $100 \times 100 \times 500$ tensors)? Are there computational bottlenecks, especially with the ODE solvers or variational inference?
>
> R2: Good question! We report the experiment with respect to the scale with the oders of tensors.  We generate synthetic dataset with different orders while fix the dimension of temporal mode to 25 and non-temporal mode to 20. We  randomly extract 1% of the total data points for training.
> Consistent with Fig. 4, we report the average training time (in seconds) per epoch.
>
>
>
> |   tensor size   |  $20 \times 20 \times 20 \times 25$ (4-order) |$20 \times 20  \times 20 \times 20 \times 25$ (5-order)| $20 \times 20  \times 20 \times 20 \times 20 \times 25$ (6-order) |
> |:-----:|:-----:|:-----:|:-----:|
> |Time per Epoch | 0.433s |1.101s|3.062s|
>
>
> CATTE scales well with the tensor order. CATTE decouples each mode and translates the addition of modes into an increase in the number of ODE states with our proposed efficent-motivated approach.
> For instance, for a 4th-order tensor of size $20 \times 20 \times 20 \times 25$, we only need to solve $20 + 20 + 20 = 60$ ODE states using three derivative networks: $h_{\theta_1}$, $h_{\theta_2}$, $h_{\theta_3}$. All the $60$ ODE states can be update simutanously with  ODE solver. Similarly, for a 6th-order tensor of size $20 \times 20 \times 20 \times 20 \times 20 \times 25$, only $20 \times 5 = 100$ ODE states need to be solved, with two additional derivative networks: $h_{\theta_4}, h_{\theta_5}$.
>
>
>
>
> Regarding scalability with respect to the spatial and temporal modes, pls refer to R8 of Reviewer q4xE.
>
> As noted in R1, our deep learning-based variational inference does not suffer from the scalability issues presented in previous methods. **Therefore, with our specific designs, the ODE solvers and variational inference do not pose  computational bottlenecks for our method.**
>
>
> We will include this clarification in the latest version
>
>
> >C3:As the author shows the robustness to different noise distributions, it is interesting how about the robustness when the data contains missing entries.
>
> R3:We would like to clarify that our method is specifically designed to complete the entire tensor from sparse observations. All experiments are conducted on datasets with a high proportion of missing entries.
>
>
> >C4:Are there any smoothness (stationary) assumptions on the temporal dynamics of the tensor? How is the performance of the CATTE if there is a sudden change?
>
> R4: We do not make explicit smoothness (or stationarity) assumptions on the temporal dynamics of the tensor. However, in the latent space,  implicit assumption on the  smoothness of latent trajectories is made  and we model them using Neural ODEs.  Visualization results of CATTE are provided in Appendix B.7. Figures 8 and 9 show that CATTE remains effective even in the presence of sudden changes.
>
>
>
>
> >C5:It seems the ref. [28] and ref. [34] are the same. Please check if there are some mistakes.
>
> R5:Thanks for the comment! We will fix them in the latest version.

---

> > ### Comment · Reviewer_MW7k · 2025-08-06
> >
> > Thank the authors for their responses. I would maintain my score.

---

> > > ### Author Response · Authors · 2025-08-06
> > >
> > > Thank you for your feedback!

---

### Official Review · Reviewer_a7FC · 2025-07-02

**Clarity:** 3
**Significance:** 3
**Originality:** 3
**Rating:** 5
**Confidence:** 3

**Summary:**

The authors propose a new temporal tensor decomposition method for tensors with continuous modes. Instead of treating time as a separate mode, their method allows latent factor functions, parametrized by neural ODEs, to depend jointly on both the mode index and continuous time. The mean-field variational posterior proposed admits analytically tractable components, allowing for efficient inference. The authors compare their method empirically with other temporal tensor decomposition methods.

**Questions:**

- L78: Brief mention of rank not having to be scalar (as in Tucker decomposition) would be helpful
- L94: I think "automatically determined" is not a good description of the authors' methodology. See my comments about this below.
- L105: "... it is inadequate for..." Why not? Please explain more concretely.
- L115: "... which requires special treatment [16]." Why? Please explain.
- L119: What is the best way to handle discrete modes with the proposed method?
- L156: Please express any potential downsides, if any, of this efficiency-motivated approach.
- L164: Please clearly state and discuss why the current modeling approach is preferred over a GP for the factor functions, making explicit the trade-offs involved.
- L199: Please direct the interested reader to the full derivations for the ELBO terms and the factor updates.
- L186: The mean-field assumption is commonly utilized in variational inference. However, given the potentially intricate interactions between factors, index variables, and time, please discuss the potential problems with this assumption. For example. would this pose a problem in modeling temporal tensors with hierarchical seasonality (months, weeks, days)?
- L248, Figure 2: Why would the uncertainty be very low in Fig 2, at the top of the second hill?
- L311: Please discuss why ELBO-based model selection is not utilized, and compare the existing method for model selection with it (in a simpler setting, if needed). You can defer the second part to camera-ready (if accepted) if there are time/compute limitations.

**Ethical Concerns:**

["NO or VERY MINOR ethics concerns only"]

**Final Justification:**

The authors provided answers for my remaining concerns, I maintain my positive score.

**Limitations:**

The authors are recommended to express the limitations of their approach taking into consideration the feedback provided above.

**Paper Formatting Concerns:**

There are no outstanding formatting concerns.

**Quality:**

3

**Strengths And Weaknesses:**

## Strengths
- The paper puts forth a clear problem, coherently expresses shortcomings of existing methods, develops a method that addresses these shortcomings, and demonstrates its favorable performance against reasonable baselines. Overall, the paper is written well.
- The solution offered addresses the original problems adequately. The neural ODE factor functions allow for expressive and smooth temporal dynamics. The probabilistic model and the proposed inference scheme are valid and appropriate, allowing for expressive yet efficient modeling.
- The empirical investigation is detailed and compares the performance of the method to important baselines. Additional experiments investigate the interpretability of the method, as well as details of rank determination

## Weakness
- Although overall the proposed method is well-motivated, arguably important modeling choices, trade-offs involved therein, and explicit justification for the choices made are not always presented in sufficient detail. The next section provides more details for this.

---

> ### Author Rebuttal · Authors · 2025-07-29
>
> Thank you for your strong support! Here are our responses. C: comments; R: response.
>
>
>
> >L78: Brief mention of rank not having to be scalar (as in Tucker decomposition) would be helpful
>
> R1: Thank you for your suggestion! We will mention it in the latest version.
>
>
> >C2:L105: "... it is inadequate for..." Why not? Please explain more concretely.
>
> R2: Good question! Model (3) incorporates temporal info by modeling continuous timestamps into latent factors (i.e., $\mathbf{u}(t)$), but it cannot handle continuity in other modes (e.g., spatial coordinates). As a result, (3) does not support learning continuous variations (like smooth changes along latitude/longitude or physical parameters) in non-temporal modes.
>
>
> >C3:L115: "... which requires special treatment [16]." Why? Please explain.
>
> R3: Great question! Model (4) is a fully factorized form, and treats all modes (including temporal) independently on an equal footing.
>
>  While this approach allows continuous indexing across all modes, **it can be overly simplistic by overlooking the fact** that the temporal mode often exhibits much sharper and more complex fluctuations than other modes (e.g, longitude, latitude). This phenomenon  is evident in the dataset we used and is also supported by the literature [16].
>
>
>
> >C4:L119: What is the best way to handle discrete modes with the proposed method?
>
> R4:  In this work, we focus on modeling physical dynamics where the mode indices are inherently continuous. For data with discrete modes, one possible extension is to incorporate ideas from tensor decomposition techniques commonly used in other domains, such as recommender systems, to adapt our model accordingly.
>
> >C5:L156: Please express any potential downsides, if any, of this efficiency-motivated approach.
>
> R5: Good point! The efficiency-driven approach is designed to enable the GPU to batch-process the states of multiple indices. This reformulation is equivalent to the original formulation, which treats each state independently. Therefore, it does not entail any loss in expressiveness or performance.
>
>
>
> >C6:L164: Please clearly state and discuss why the current modeling approach is preferred over a GP for the factor functions, making explicit the trade-offs involved.
>
> R6:  Good question! Neural ODEs have several advangtages over GPs in terms of scalability, flexibility, expressiveness.
>
> * GPs have a computational complexity of $O(N^3)$. To make it practical, various techniques, such as sparsification and conversion to state-space models (as done in FunBaT), are used, further complicating implementation. In contrast, Neural ODEs scale linearly with the number of data, making them a more straightforward choice for large-scale problems.
>
> * Neural ODEs and GPs are both strong approaches for modeling dynamics. However, the expressiveness of GPs heavily depends on the choice of kernels and hyperparameters, which often requires domain expertise and makes hyperparameter optimization challenging. In contrast, Neural ODEs can achieve high expressiveness more easily by simply using MLPs.
>
>
> One key advantage of Gaussian Processes is their ability to provide uncertainty quantification—a feature that is absent in standard neural ODEs. However, this gap is bridged in our proposed framework, which incorporates uncertainty modeling within a principled probabilistic setting.
>
> >C7:L199: Please direct the interested reader to the full derivations for the ELBO terms and the factor updates.
>
> R7: Thank you for the suggestion. In the revised manuscript, we will direct interested readers to Appendix A, which contains the full derivations of the ELBO terms and the corresponding factor updates.
>
>
> >C8:L186: The mean-field assumption is commonly utilized in variational inference. However, given the potentially intricate interactions between factors, index variables, and time, please discuss the potential problems with this assumption. For example. would this pose a problem in modeling temporal tensors with hierarchical seasonality (months, weeks, days)?
>
> R8:  Good question! We agree that the mean-field assumption simplifies variational inference by assuming independence among latent variables, which can lead to suboptimal posterior estimates. However, in our model, we leverage neural networks to model the posterior means of the latent functions, thereby enhancing their representatitive capacity. The incorporated neural ODEs are effective at capturing complex temporal patterns, such as hierarchical seasonality. Extensive experiments confirm that our approach handles complex temporal tensors well. For empirical evidence, please refer to Appendix B.7, which includes results on datasets with rich seasonal structures, where our method performs robustly.
>
>
>
> >C9:L248, Figure 2: Why would the uncertainty be very low in Fig 2, at the top of the second hill?
>
> R9: Fig.2(b) shows the temporal trajectory indexed at $(0.992, 0.982)$. The peak of the second hill does not contain any training points. However, in a very near trajectory (i.e., at $(0.986, 0.982)$), some training points are observed around the second hill. Our model successfully captures the continuity of the underlying function, resulting in a correspondingly low uncertainty in this region.
>
> >C10:L311: Please discuss why ELBO-based model selection is not utilized, and compare the existing method for model selection with it (in a simpler setting, if needed). You can defer the second part to camera-ready (if accepted) if there are time/compute limitations.
>
>
> R10: Thank you for the suggestion. ELBO-based model selection involves multiple trials, which can be time-consuming. We will compare the existing method for model selection in the latest version.

---

> > ### Comment · Reviewer_a7FC · 2025-08-04
> >
> > I thank the authors for their clarifications and maintain my score.

---

> > > ### Author Response · Authors · 2025-08-04
> > >
> > > Thank you once again for your  support！

---

### Decision · Program_Chairs · 2025-09-17

**Decision:**

Accept (poster)

**Comment:**

This paper introduces Catte, a functional complexity-adaptive temporal tensor decomposition method. Unlike prior temporal tensor models that only handle continuous timestamps, Catte generalizes to tensors with continuous indexes across multiple modes, such as spatial coordinates in climate data. The method uses learnable Fourier features to encode continuous spatial inputs and neural ODEs to model temporal factor trajectories. To automatically adapt model complexity, it applies a sparsity-inducing prior over trajectories, coupled with an efficient variational inference scheme that provides a closed-form evidence lower bound without sampling.

The paper is technically solid, though the methodology is a bit complex for practical applications. The design choices are overall appropriate. The reviewers appreciate the linear scaling. While all individual components (Neural ODEs, Bayesian rank selection, functional tensors) exist and the novelty might look a bit incremental, as long as the proposed methodology is technically solid, I am fine with that.